# Multi-Resolution Decomposable Diffusion Model for Non-stationary Time Series Anomaly Detection

**Guojin Zhong**[1*]   **Pan Wang**[1*]   **Jin Yuan**[1†]   **Zhiyong Li**[1]   **Long Chen**[2]
[1]Hunan University   [2]Hong Kong University of Science and Technology
{gjzhong, yuanjin, zhiyong.li}@hnu.edu.cn   longchen@cse.ust.hk

## Abstract

Recently, generative models have shown considerable promise in unsupervised time series anomaly detection. Nonetheless, the task of effectively capturing complex temporal patterns and minimizing false alarms becomes increasingly challenging when dealing with non-stationary time series, characterized by continuously fluctuating statistical attributes and joint distributions. To confront these challenges, we underscore the benefits of multi-resolution modeling, which improves the ability to distinguish between anomalies and non-stationary behaviors by leveraging correlations across various resolution scales. In response, we introduce a **M**ulti-Res**o**lution **De**composable Diffusion **M**odel (MODEM), which integrates a coarse-to-fine diffusion paradigm with a frequency-enhanced decomposable network to adeptly navigate the intricacies of non-stationarity. Technically, the coarse-to-fine diffusion model embeds cross-resolution correlations into the forward process to optimize diffusion transitions mathematically. It then innovatively employs low-resolution recovery to guide the reverse trajectories of high-resolution series in a coarse-to-fine manner, enhancing the model's ability to learn and elucidate underlying temporal patterns. Furthermore, the frequency-enhanced decomposable network operates in the frequency domain to extract globally shared time-invariant information and time-variant temporal dynamics for accurate series reconstruction. Extensive experiments conducted across five real-world datasets demonstrate that our proposed MODEM achieves state-of-the-art performance and can be generalized to other time series tasks.

## 1 Introduction

Time series anomaly detection has emerged as a critical task in various domains such as industrial manufacturing, finance, and healthcare monitoring, aiming to identify anomalous data within specific time intervals (Xu et al., 2022; Chen et al., 2023; Yang et al., 2023). Considering the scarcity of anomaly labels, most existing time series anomaly detection methods operate under an unsupervised paradigm, trained solely on data with normal behavior. These methods are generally divided into three categories: density-based, forecasting-based and reconstruction-based. Density-based methods (Dai & Chen; Zhou et al., 2023; 2024) detect anomalies based on the assumption that anomalies often lie on low-density regions of data distribution. Forecasting-based methods (Zong et al., 2018; Audibert et al., 2020; Yao et al., 2022) predict future values from past observations, while reconstruction-based methods (Su et al., 2019; Chen et al., 2023; Xiao et al., 2023) aim to reconstruct the entire input series. Both types identify anomalies by comparing the errors between the predicted or reconstructed series and the actual input series, with large discrepancies indicating potential anomalies (Paparrizos et al., 2022; Li et al., 2023b). Despite their effectiveness, these methods are primarily optimized for stable time series and are prone to false alarms when dealing with normal yet non-stationary data because non-stationary time series continuously change their statistical properties and joint distribution over time (as shown in the gray area of Fig. 1(a)). Consequently, these methods struggle to accurately

---

*Contributed equally.
†Corresponding authors.

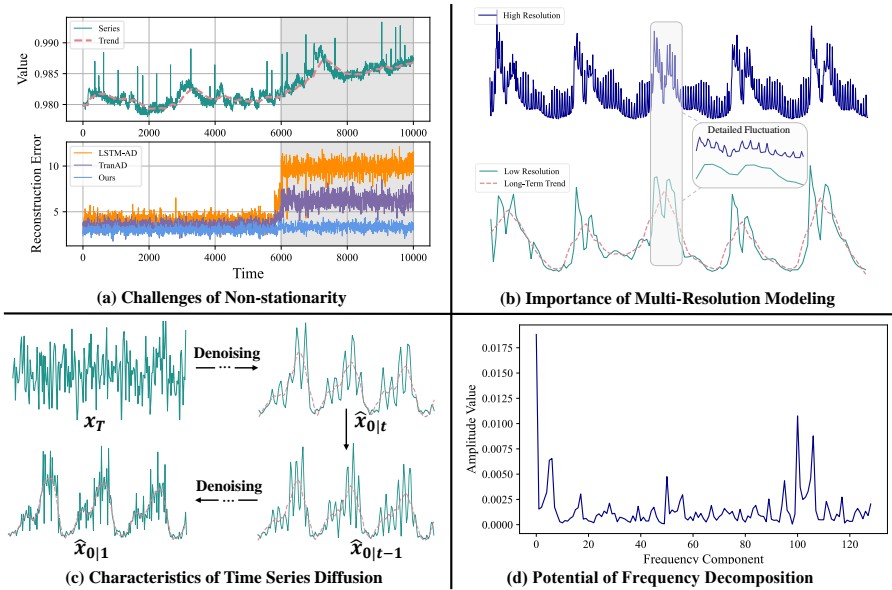

Figure 1: Several examples to illustrate the challenges of non-stationarity and the motivation of the use of multi-resolution modeling and frequency decomposition.

model non-stationarity, often resulting in high reconstruction errors that are mistakenly classified as anomalies (as illustrated by the orange and purple curves in Fig. 1(a)).

To address non-stationary time series in anomaly detection, $D^3R$ (Wang et al., 2024) employs a moving average kernel to extract labeled stable components, which then guide the decomposition of stable and trend components for training the diffusion model. However, $D^3R$ is designed for single-resolution data and lacks the capability to effectively utilize multi-scale temporal information to detect anomalies in non-stationary time series. Moreover, its reliance on the moving average strategy, which presupposes a linear trend, is often inadequate for handling non-stationary time series characterized by complex seasonal variations or multiple temporal patterns. Consequently, this approach may even introduce erroneous lag signals, potentially compromising the detection accuracy.

To enhance the robustness of anomaly detection, we emphasize the significance of multi-resolution modeling in capturing the underlying non-stationary patterns. Low-resolution time series typically display prominent long-term trends, while high-resolution series provide insights into detailed fluctuations and short-term changes, as shown in Fig. 1(b). By exploiting correlations across multiple resolution scales, the model gains a comprehensive view of data, benefiting from both overarching trend information and granular event details. Furthermore, as depicted in Fig. 1(c), time series diffusion models (Li et al., 2022b; Shen et al., 2024) typically generate the overall trend in the early denoising steps and then refine local temporal patterns in the later steps. This observation intuitively guides our strategy of sequentially reconstructing low-resolution data followed by high-resolution data using cross-resolution correlations, thereby establishing a more accurate reconstruction baseline for anomaly detection.

Motivated by these insights, we propose a **M**ulti-Res**o**lution **De**composable Diffusion **M**odel (MO-DEM) to enhance anomaly detection in non-stationary time series. MODEM leverages intrinsic correlations across different resolution scales to inform the learning process of the diffusion model. It explicitly incorporates the cross-scale correlations between high-resolution and low-resolution data into the forward process, serving as the prior guidance to optimize the diffusion transitions. During denoising process, these cross-scale correlations are harnessed to adapt the reverse trajectories, where predictions at lower resolutions aid in the coarse-to-fine recovery of higher-resolution data. Additionally, we have extended MODEM to the Denoising Diffusion Implicit Models (DDIM) to accelerate the sampling process. To effectively parameterize the denoising process, we design a frequency-enhanced decomposable network, which employs the short-term Fourier transform to analyze the input series. An examination of the spectrum reveals marked disparities in amplitude energy across various frequency components (Fig. 1(d)). Higher amplitude levels point to globally dominant time-invariant components, whereas lower amplitudes correspond to time-variant patterns.

Drawing on these insights, non-stationary series are decomposed into time-invariant and time-variant components. Subsequently, two distinct encoders are deployed to separately extract globally shared information and complex temporal dynamics, thereby enhancing the modeling of non-stationarity. Extensive experiments conducted on five real-world datasets demonstrate that MODEM significantly surpasses state-of-the-art (SOTA) unsupervised anomaly detection methods and exhibits strong generalization capabilities across various time series tasks. Our main contributions are listed below:

- We introduce MODEM, a diffusion model that investigates correlations across different resolution scales for non-stationary time series anomaly detection. For the first time, we explore enhancing the diffusion model's understanding of time-varying properties induced by non-stationarity from diverse temporal resolution perspectives.

- We leverage cross-resolution correlations as guidance to optimize both forward and backward trajectories, synthesizing time series in a low-to-high resolution manner. This approach, grounded in mathematical derivations, not only provides more robust reconstruction signals for anomaly detection but also exhibits strong generalization across various tasks.

- We propose a frequency-enhanced decomposable network to effectively parameterize the denoising process. This network operates in the frequency domain to decompose time-invariant and time-variant components, allowing for the extraction of globally shared information and the identification of underlying temporal dynamics.

## 2 RELATED WORK

Existing unsupervised anomaly detection methods for time series are primarily categorized into density-based (Dai & Chen; Zhou et al., 2023), forecasting-based (Yao et al., 2022), and reconstruction-based (Zong et al., 2018; Audibert et al., 2020) approaches. Density-based methods focus on fitting the density of training and test samples to detect anomalies, which are based on the assumption that anomalies often lie on low-density regions of data distribution (Zhou et al., 2024). Forecasting-based methods utilize past series data to predict future values through various modified neural networks, including Long Short-Term Memory (LSTM) (Hundman et al., 2018), graph neural networks (Deng & Hooi, 2021; Zhao et al., 2020a), and Generative Adversarial Networks (GAN) (Yao et al., 2022; Zhong et al., 2023a). These methods detect anomalies based on the forecasting error between the predicted and actual values, but they are prone to interference from historical data. In contrast, reconstruction-based methods involve reconstructing the entire input series and identifying anomalies based on the reconstruction error. Early methods in this category use Variational Auto-Encoders (VAE) (Su et al., 2019) or GAN (Li et al., 2019) to reconstruct time series. Inspired by the success of transformers (Liu et al., 2023; Chen et al., 2025), TranAD (Tuli et al., 2022) introduces the self-attention mechanism to learn temporal patterns and incorporates adversarial training to enhance model robustness. Anomaly Transformer (Xu et al., 2022) utilizes the proximity concentration deviation of anomalies to make rare exceptions more distinguishable. However, the training instability and significant error accumulation in transformers have been challenged by diffusion models (Ho et al., 2020; Zhong et al., 2024; 2025), which recently demonstrate superior performance in time series anomaly detection. For instance, ImDiffusion (Chen et al., 2023) employs a diffusion model to fill the masked time series and proposes an ensemble strategy for detecting anomalous data. Similarly, DiffAD (Xiao et al., 2023) adopts the mask-then-imputation paradigm and designs a conditional weight-incremental diffusion model to improve the interpolation performance of missing series. Due to the drift caused by non-stationary environments, $D^3R$ (Wang et al., 2024) dynamically decomposes stable and trend components using the mix-attention technique and leverages the diffusion model to reconstruct the input series, achieving significant performance improvements.

Notably, the importance of multi-resolution techniques Li et al. (2023a); Zhang et al. (2024) has been demonstrated in time series forecasting. For instance, MG-TSD Fan et al. (2024) designs a multi-granularity guidance loss to guide the learning process of diffusion models, while MR-Diff Shen et al. (2024) employs progressive denoising to generate coarser and finer trend signals, improving the accuracy of time series prediction. These works inspire us to be the first to analyze normal and anomalous change patterns in non-stationary data across multiple resolution scales.

In this work, we further explore the potential of diffusion models for detecting anomalies in non-stationary time series. Uniquely, we extend the diffusion model into a multi-resolution paradigm,

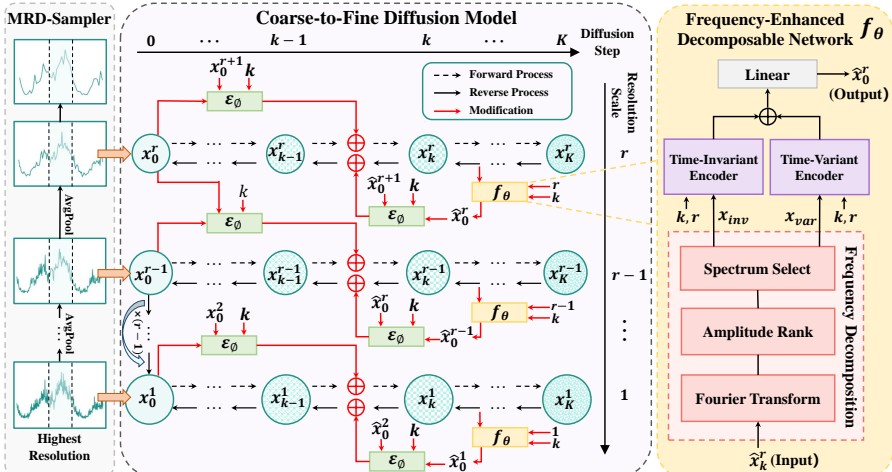

Figure 2: Overview of Multi-Resolution Decomposable Diffusion Model, consisting of three modules: a Multi-Resolution Data Sampler (MRD-Sampler), a Coarse-to-Fine Diffusion Model, and a Frequency-Enhanced Decomposable Network for non-stationary time series anomaly detection.

which explicitly utilizes correlations within different time scales to optimize the forward and reverse trajectories, effectively alleviating the interference of non-stationarity on anomaly detection.

## 3 METHODOLOGY

### 3.1 PROBLEM STATEMENT

Let $\mathbf{X} \in \mathbb{R}^{N \times T}$ denotes observed time series, where $N$ and $T$ are number of variables and length of timesteps, respectively. The anomaly detection task aims to produce an output vector $\mathbf{Y} \in \mathbb{R}^T$ based on input series $\mathbf{X}$, where $y_i = 0$ or $1$ indicates whether the $i$-th timestep is an anomaly. The temporal patterns of non-stationary time series typically change over time because they originate from dynamic environments, which poses challenges for anomaly detection.

### 3.2 OVERVIEW OF MODEM ARCHITECTURE

The architecture of MODEM is depicted in Fig. 2, which mainly consists of a multi-resolution data sampler, a coarse-to-fine diffusion model, and a frequency-enhanced decomposable network.

**Multi-Resolution Data Sampler**   Given a time series $\mathbf{x}_0$ and a resolution scale $r \in [1, R]$, this step employs average pooling (Wu et al., 2021) to generate multi-resolution data, $\mathbf{x}_0^r$, that is, $\mathbf{x}_0^r = \text{AvgPool}(\text{Padding}(\mathbf{x}_0), 2^r)$, where $2^r$ denotes the pooling size that increases with $r$. $\text{AvgPool}(\cdot)$ computes the average of every $2^r$ non-overlapping points. $\text{Padding}(\cdot)$ is then applied by replicating the pooled data at $r$ resolution $2^r$ times to maintain the same length as the highest-resolution series $\mathbf{x}_0^1$. This strategy provides the necessary data support for our model's subsequent multi-resolution learning, enabling it to capture evolutionary characteristics of non-stationary time series across multiple resolutions and enhancing its understanding of normal and anomalous change patterns.

**Coarse-to-Fine Diffusion Model**   Multi-resolution data $\{\mathbf{x}_0^R, \mathbf{x}_0^{R-1}, ..., \mathbf{x}_0^1\}$ are input into a coarse-to-fine diffusion model. Different from the existing diffusion models, they are sequentially disturbed as $\{\mathbf{x}_K^R, \mathbf{x}_K^{R-1}, ..., \mathbf{x}_K^1\}$ from low to high resolution in $R \times K$ forward steps. During the reverse process, noisy samples are progressively restored to their original form in a coarse-to-fine manner, with the clean prediction $\hat{\mathbf{x}}_0^r$ aiding in the denoising of the higher-resolution data $\mathbf{x}_k^{r-1}$. A detailed discussion of this model can be found in Section 3.3.

**Frequency-Enhanced Decomposable Network**   It is designed to parameterize the denoising process of the diffusion model, enabling the prediction of clean data at each diffusion step $k$. It

comprises three modules specifically tailored to learn about the underlying pattern of non-stationarity. For a detailed description of this design, please refer to Section 3.4.

## 3.3 COARSE-TO-FINE DIFFUSION MODEL

Motivated by (Fan et al., 2024), we observe that the forward process of the diffusion model intuitively mirrors the smoothing of time series from high to low resolution. Additionally, upon revisiting the conventional denoising process, we find that the diffusion model tends to reconstruct the overall sequence first and then refine the local temporal fluctuations, as illustrated in Fig. 1(c). These insights lead us to extend the time series diffusion model to a multi-resolution paradigm that utilizes the intrinsic correlations within different resolution scales as prior guidance for improved performance.

**Multi-Resolution Forward Process** Rather than naively adding random noises on all $R$ resolutions $\{\mathbf{x}_0^1, \mathbf{x}_0^2, ..., \mathbf{x}_0^R\}$ at each of the $K$ diffusion steps, we execute the diffusion process sequentially from the lowest-resolution $\mathbf{x}_0^R$ to the highest-resolution $\mathbf{x}_0^1$. Here, each scale undergoes $K$ diffusion steps, modifying the total number of diffusion steps to $R \times K$. Furthermore, we strive to incorporate the cross-scale correlation between $\mathbf{x}_0^r$ and $\mathbf{x}_0^{r+1}$ into the forward process. This ensures the diffusion model maintains the underlying normal pattern from $\mathbf{x}_0^{r+1}$ to guide the synthesis of the higher-resolution time series $\mathbf{x}_0^r$. Specifically, $\mathbf{x}_0^r$ and $\mathbf{x}_0^{r+1}$ are input into a transformer-based module designed to learn their cross-scale correlation through cross-attention (Yang et al., 2024). This cross-scale correlation can intuitively serve as prior guidance to adaptively optimize diffusion trajectories across all steps:

$$q_\phi(\mathbf{x}_k^r|\mathbf{x}_0^r, \mathbf{x}_0^{r+1}) = \mathcal{N}(\mathbf{x}_k^r, \sqrt{\bar{\alpha}_k}\mathbf{x}_0^r + \gamma_k \mathcal{E}_\phi(\mathbf{x}_0^r, \mathbf{x}_0^{r+1}, k), (1 - \bar{\alpha}_k)\mathbf{I}), \tag{1}$$

where $\gamma_k = \sqrt{\bar{\alpha}_k} \cdot (1 - \sqrt{\bar{\alpha}_k})$ is a step-varying factor controlling the intensity of prior information. $\mathcal{E}_\phi$ denotes the lightweight module with parameter $\phi$. Inspired by, but distinct from, ShiftDDPM Zhang et al. (2023) and ContextDiff Yang et al. (2024), which utilize classifiers and cross-modal contextual information respectively, we are the first to leverage the temporal correlations across multiple resolution scales to modify the diffusion trajectory, ensuring that lower-resolution data effectively facilitate the reconstruction of higher-resolution data. Subsequently, we can derive the multi-resolution forward transition given $\mathbf{x}_{k-1}^r, \mathbf{x}_0^r$, and $\mathbf{x}_0^{r+1}$ (see detailed proof in Appendix A.1):

$$q_\phi(\mathbf{x}_k^r|\mathbf{x}_{k-1}^r, \mathbf{x}_0^r, \mathbf{x}_0^{r+1}) = \mathcal{N}(\sqrt{\alpha_k}\mathbf{x}_{k-1}^r + \gamma_k \mathcal{E}_\phi(\mathbf{x}_0^r, \mathbf{x}_0^{r+1}, k) - \sqrt{\alpha_k}\gamma_{k-1}\mathcal{E}_\phi(\mathbf{x}_0^r, \mathbf{x}_0^{r+1}, k-1), \beta_k\mathbf{I}). \tag{2}$$

At each step $k$, the model explicitly considers cross-scale correlation to optimize the forward transitions, ensuring a more effective alignment with the coarse-to-fine denoising process. Using Eq. (1) and Eq. (2), we can further derive the posterior distribution of the forward process for $k > 1$ by applying Bayes' rule (see detailed proof in Appendix A.2):

$$\begin{aligned} q_\phi(\mathbf{x}_{k-1}^r|\mathbf{x}_k^r, \mathbf{x}_0^r, \mathbf{x}_0^{r+1}) = &\mathcal{N}(\frac{\sqrt{\bar{\alpha}_{k-1}}\beta_k}{1 - \bar{\alpha}_k}\mathbf{x}_0^r + \frac{\sqrt{\alpha_k}(1 - \bar{\alpha}_{k-1})}{1 - \bar{\alpha}_k}(\mathbf{x}_k^r - \gamma_k \mathcal{E}_\phi(\mathbf{x}_0^r, \mathbf{x}_0^{r+1}, k)) \\ &+ \gamma_{k-1}\mathcal{E}_\phi(\mathbf{x}_0^r, \mathbf{x}_0^{r+1}, k-1), \frac{(1 - \bar{\alpha}_{k-1})\beta_k}{1 - \bar{\alpha}_k}\mathbf{I}). \end{aligned} \tag{3}$$

In this manner, the forward process of MODEM begins with the corruption of low-resolution time series, which display long-term trends and periodic variations. This is followed by the degradation of high-resolution time series that preserve short-term fluctuations. These diffusion transitions are steered by cross-scale correlations, enabling MODEM to seamlessly integrate and capture both macroscopic and microscopic details of non-stationary time series across different scales.

**Parameterized Reverse Process** To align with the aforementioned forward process, the reverse phase of MODEM aims to restore multi-resolution noisy data $\{\mathbf{x}_K^R, \mathbf{x}_K^{R-1}, ..., \mathbf{x}_K^1\}$ back to clean data $\{\mathbf{x}_0^R, \mathbf{x}_0^{R-1}, ..., \mathbf{x}_0^1\}$. Without loss of generality, we revisit the variational bound of $\log p(\mathbf{x}_0^1)$, and seek parameterized kernels $p_\theta(\mathbf{x}_{k-1}^r|\mathbf{x}_k^r, \mathbf{x}_0^{r+1}) = \mathcal{N}(\mu_\theta, \Sigma_\theta)$ to approximate $q_\phi(\mathbf{x}_{k-1}^r|\mathbf{x}_k^r, \mathbf{x}_0^r, \mathbf{x}_0^{r+1})$ in Eq. (3) by minimizing their KL-divergence (Zhong et al., 2023b) $D_{KL}(q_\phi(\mathbf{x}_{k-1}^r|\mathbf{x}_k^r, \mathbf{x}_0^r, \mathbf{x}_0^{r+1})\|p_\theta(\mathbf{x}_{k-1}^r|\mathbf{x}_k^r, \mathbf{x}_0^{r+1}))$. By combining Eq. (1) and Eq. (3), and assuming that $\Sigma_\theta$ equals the posterior variance $\frac{(1 - \bar{\alpha}_{k-1})\beta_k}{1 - \bar{\alpha}_k}$, we can derive the distribution of the

parameterized kernels (see detailed proof in Appendix A.3):

$$
\begin{aligned}
p_\theta(\mathbf{x}_{k-1}^r | \mathbf{x}_k^r, \mathbf{x}_0^{r+1}) = \mathcal{N}(&\frac{1}{\sqrt{\alpha_k}}\left[\mathbf{x}_k^r - \frac{\beta_k}{\sqrt{1-\bar{\alpha}_k}}f_\theta(\mathbf{x}_k^r, k, r)\right] - \frac{\sqrt{\alpha_k(1-\bar{\alpha}_{k-1})}}{1-\bar{\alpha}_k}\gamma_k \mathcal{E}_\phi(\mathbf{x}_0^r, \mathbf{x}_0^{r+1}, k) \\
&+ \gamma_{k-1}\mathcal{E}_\phi(\mathbf{x}_0^r, \mathbf{x}_0^{r+1}, k-1), \frac{(1-\bar{\alpha}_{k-1})\beta_k}{1-\bar{\alpha}_k}\mathbf{I}),
\end{aligned}
\tag{4}
$$

where $f_\theta(\mathbf{x}_k^r, k, r)$ denotes the denoising network with parameter $\theta$ that predicts the noise. Thus the KL-divergence objective can be equivalent to:

$$
\mathcal{L}_{\theta,\phi,k,r} = \sum_{k=1}^K \eta_k \mathbb{E}_{\mathbf{x}_0^r,,r}\left[\|f_\theta(\mathbf{x}_k^r, k, r) - \frac{\mathbf{x}_k^r - \sqrt{\bar{\alpha}_k}\mathbf{x}_0^r}{\sqrt{1-\bar{\alpha}_k}}\|^2\right],
\tag{5}
$$

where $\mathbf{x}_k^r = \sqrt{\bar{\alpha}_k}\mathbf{x}_0^r + \gamma_k \mathcal{E}_\phi(\mathbf{x}_0^r, \mathbf{x}_0^{r+1}, k) + \sqrt{1-\bar{\alpha}_k}$ and $\eta_k = \frac{\beta_k}{2\alpha_k(1-\bar{\alpha}_{k-1})}$ is a loss weight.

**Accelerated Training and Sampling**   To enhance the efficiency of MODEM, we accelerate both the training and sampling procedures. Firstly, it is evident from Eq. (5) that $\mathbf{x}_k^r$ introduces additional cross-scale correlation calculations at each training step. To mitigate this, our denoising network $f_\theta(\mathbf{x}_k^r, k, r)$ is designed to directly predict the clean data $\mathbf{x}_0^r$ at resolution $r$ from $\mathbf{x}_k^r$, rather than estimating the noise. The refined training objective is defined as follows:

$$
\mathcal{L}_{diff} = \sum_{k=1}^K \mathbb{E}_{\mathbf{x}_0^r, r}\left[\|f_\theta(\mathbf{x}_k^r, k, r) - \mathbf{x}_0^r\|^2\right].
\tag{6}
$$

Secondly, modeling time series at $R$ different resolutions leads to inefficiencies, as the number of sampling steps increases by a factor of $R$. Drawing inspiration from DDIM (Song et al., 2020), the reverse process of MODEM is adapted to be deterministic and we can obtain $\mathbf{x}_{k-1}^r$ given $\mathbf{x}_k^r$ via (see detailed implementations in Appendix A.4):

$$
\begin{aligned}
\mathbf{x}_{k-1}^r = &\sqrt{\bar{\alpha}_{k-1}}\hat{\mathbf{x}}_0^r + \sqrt{1-\bar{\alpha}_{k-1}-\sigma_k^2} \cdot \frac{\mathbf{x}_k^r - \sqrt{\bar{\alpha}_k}\hat{\mathbf{x}}_0^r}{\sqrt{1-\bar{\alpha}_k}} \\
&- \gamma_k \mathcal{E}_\phi(\hat{\mathbf{x}}_0^r, \hat{\mathbf{x}}_0^{r+1}, k) \cdot \frac{1-\bar{\alpha}_{k-1}-\sigma_k^2}{\sqrt{1-\bar{\alpha}_k}} + \gamma_{k-1}\mathcal{E}_\phi(\hat{\mathbf{x}}_0^r, \hat{\mathbf{x}}_0^{r+1}, k-1).
\end{aligned}
\tag{7}
$$

Thus, we can employ a sub-sequence of $[1, ..., K]$ of length $L$, where $L \ll K$. As a result, the total number of denoising steps is reduced from $R \times K$ to $R \times L$, resulting in a faster sampling. In contrast to the conventional reverse process, MODEM utilizes coarser-grained temporal patterns to adaptively constrain the sampling trajectories of finer-grained time series, thereby effectively reducing the interference caused by non-stationarity and produces precise normal baselines for anomaly detection.

### 3.4   FREQUENCY-ENHANCED DECOMPOSABLE NETWORK

To parameterize the denoising process of MODEM, we design a Frequency-Enhanced Decomposable Network that learns the underlying dynamics of non-stationarity. The proposed network incorporates a frequency decomposition module to disentangle time-invariant and time-variant components. Following this, two distinct encoders are employed to process these components separately.

**Frequency Decomposition**   Previous works (Yang & Hong, 2022; He et al., 2023) have demonstrated that spectral responses can more robustly capture underlying temporal patterns compared to time-domain representations. Motivated by these insights, we can utilize spectral statistics to disentangle the time series components: Time-invariant components, which represent dominant stationary factors, typically manifest as high-magnitude frequency components when transformed into the frequency domain, while time-variant components reflecting non-stationarity are characterized by low-magnitude frequency components. Specifically, we employ the short-term Fourier transform (STFT), which is well-suited for time-varying signals, to compute the mean amplitude for each spectrum, denoted as $\mathcal{A} = \{0, 1, ..., \frac{T}{2}\}$. We then select the top $m$ percent of the highest amplitudes as a subset $\mathcal{A}_m$, which captures the dominant time-invariant information (Liu et al., 2024). The disentanglement is performed as follows:

$$
\mathbf{x}_{inv} = \mathcal{F}^{-1}(\text{Select}(\mathcal{A}_m, \mathcal{F}(\mathbf{x}))), \quad \mathbf{x}_{var} = \mathbf{x} - \mathbf{x}_{inv},
\tag{8}
$$

where $\mathcal{F}$ denotes STFT and $\mathcal{F}^{-1}$ is its inverse transformation, Select($\cdot$) allows only the selected frequency components to pass through.

**Time-Invariant Encoder**  Given the time-invariant component $\mathbf{x}_{inv}$, we utilize several stacked residual blocks to encode the globally shared information. Recognizing that understanding the relationships between different variables is advantageous for modeling non-stationarity (Tashiro et al., 2021; Liu et al., 2023), each residual block includes hierarchical transformers (Chen et al., 2023) to capture both intra-series temporal correlations and inter-series interactions among variables (see Appendix B.5 for detailed architecture).

**Time-Variant Encoder**  To model underlying temporal dynamics, we develop the time-variant encoder to process $\mathbf{x}_{var}$. As illustrated in Fig. 3, it comprises multiple Dilated ModernTCN Blocks (DMTB). DMTB is based on the ModernTCN module (Luo & Wang, 2024), which connects a 1D depthwise convolutional layer (DWConv) and a convolutional feed forward network (ConvFFN) in series, demonstrating excellent performance in time series tasks. To uncover the complex periodic characteristics of $\mathbf{x}_{var}$, we expand the kernel size of DWConv using a dilated factor $d$, forming $c$ parallel ModernTCN modules. These convolution kernels of varying sizes, with different receptive fields, slide across multiple time scales, effectively capturing periodic variations of different magnitudes. Notably, they share the same parameters without introducing additional parameters. Subsequently, we utilize additive concatenation (Hu et al., 2023) to fuse their output results.

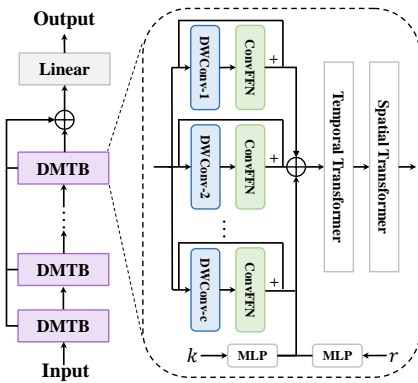

Figure 3: Architecture of Time-Variant Encoder, consisting of multiple stacked Dilated ModernTCN Blocks (DMTB).

Moreover, diffusion step and resolution-scale embeddings are added as complementary information and hierarchical transformers are incorporated to enhance temporal representation similar to the time-invariant encoder.

# 4 EXPERIMENTS

## 4.1 DATASETS AND EVALUATION METRICS

**Datasets.**  We evaluate the performance of MODEM on five real-world datasets: SMD (Server Machine Dataset) (Su et al., 2019), PSM (Pooled Server Metrics) (Abdulaal et al., 2021), MSL (Mars Science Laboratory) (Hundman et al., 2018), SWaT (Secure Water Treatment) (Mathur & Tippenhauer, 2016), and SMAP (Soil Moisture Active Passive satellite) (Entekhabi et al., 2010).

**Evaluation Metrics.**  For performance evaluation, we employ Precision (P), Recall (R), and F1-score metrics (Deng & Hooi, 2021; Zhao et al., 2020b). Additionally, we introduce the Average Sequence Detection Delay (ADD) metric (Tuli et al., 2022) to evaluate the speed and timeliness of each time series anomaly detection algorithm.

## 4.2 IMPLEMENTATION DETAILS

The proposed MODEM is implemented using the PyTorch framework and optimized using the Adam optimizer (Kingma & Ba, 2014) with a learning rate of $1e-3$ and a weight decay rate of $1e-6$. We train MODEM for 100 epochs on datasets that contain only normal time series on four NVIDIA A6000 GPUs. For anomaly detection, we utilize the ensemble inference strategy (Chen et al., 2023) to improve the robustness and accuracy of MODEM. Specifically, this involves integrating the denoising results at each diffusion step and resolution scale to vote on whether a point is anomalous (see Appendix B.4 for detailed explanation), and we set the voting threshold to 10 for all datasets. Our method employs a square noise schedule, uses 50 diffusion steps, and operates across 4 resolution scales. Further details on the hyperparameters of MODEM can be found in Appendix B.6.

## 4.3 DETECTION RESULTS

The detection performance of our MODEM is presented alongside the baseline methods in Tab. 1, where F1* denotes the average F1-score. Notably, these metrics are calculated using the point

adjustment strategy (Xu et al., 2022), in line with the approach used in most previous studies (Deng & Hooi, 2021; Zhao et al., 2020b). Additionally, the timeliness of these methods is detailed in Tab. 2, and the comparison of model efficiency is illustrated in Tab. 14 of Appendix B.10. Based on these results, we can draw the following conclusions:

Firstly, thanks to the optimized diffusion and denoising trajectories in multi-resolution time series and the decomposition of time-variant components in the frequency domain, our MODEM effectively identifies anomalous data. This is evidenced by its consistently superior performance in four out of the five datasets, with an average F1-score improvement of 2.11% (from 91.35% to 93.46%). The only exception is the MSL dataset, where MODEM is slightly less effective compared to TranAD which focuses more on exploring inter-variable correlations, better matching the characteristics of MSL. Additionally, on the SMD and SWaT datasets, known for their high non-stationarity, our approach achieves significant performance improvements. This suggests that MODEM can effectively learn underlying temporal dynamics and mitigate the impact of non-stationarity on anomaly detection.

Secondly, anomaly detection methods that utilize generative models, such as VAEs and diffusion, to reconstruct the time series consistently outperform prediction methods like LSTM-based or GNN-based methods. This indicates that generative models with robust reconstruction capabilities may provide a more effective paradigm for detecting anomalies in non-stationary time series.

Thirdly, benefiting from both the general trend information provided by low-resolution data and the detailed event information from high-resolution data, our model is able to detect anomalies more promptly, resulting in a lower ADD. Moreover, despite processing time series at multiple resolution scales, our MODEM still demonstrates competitive computational efficiency. This is attributed to the mathematical extension of the multi-resolution diffusion process to the DDIM paradigm, which enables MODEM to sample a sub-sequence at each resolution, significantly accelerating the sampling process. Further discussions on computational efficiency have been included in Appendix B.10.

Table 1: Comparison of detection performance across five real-world datasets using point adjustment strategy. The best and second-best performances are highlighted in bold and underlined, respectively.

| Method | SMD | | | PSM | | | MSL | | | SWaT | | | SMAP | | | F1* |
|---|---|---|---|---|---|---|---|---|---|---|---|---|---|---|---|---|
| | P | R | F1 | P | R | F1 | P | R | F1 | P | R | F1 | P | R | F1 | |
| IForest | 20.30 | 21.30 | 17.99 | 66.30 | 49.19 | 56.41 | 60.59 | 53.28 | 53.34 | 97.64 | 66.50 | 79.07 | 28.86 | 76.71 | 41.63 | 49.69 |
| LSTM-AD | 33.61 | 32.29 | 26.39 | 90.50 | 77.07 | 83.13 | 73.30 | 57.45 | 63.78 | 99.25 | 67.37 | 80.26 | 78.41 | 56.30 | 65.33 | 63.78 |
| MSCRED | 85.67 | 90.38 | 84.26 | 95.55 | 68.57 | 79.65 | 50.08 | 60.88 | 48.99 | 84.23 | 40.66 | 55.48 | 41.07 | 86.04 | 27.12 | 59.10 |
| GDN | 84.60 | 78.62 | 78.65 | 87.50 | 83.85 | 85.64 | 86.68 | 80.27 | 83.42 | 13.11 | 5.85 | 8.08 | 96.89 | 54.01 | 69.36 | 65.03 |
| MTAD-GAT | 88.36 | 83.30 | 84.63 | 87.63 | 87.25 | 87.44 | 84.68 | 82.24 | 83.44 | 84.68 | 82.24 | 83.44 | 97.18 | 52.59 | 68.24 | 81.44 |
| OmniAnomaly | 87.51 | 90.52 | 87.75 | 92.55 | 95.51 | 91.11 | 83.21 | 82.15 | 82.68 | 97.49 | 75.00 | 84.70 | 84.07 | 96.74 | 89.95 | 87.24 |
| InterFusion | 88.15 | 90.71 | 87.72 | 91.28 | 93.26 | 92.26 | 76.88 | 94.64 | 84.42 | 96.83 | 85.30 | 90.60 | 87.88 | 77.04 | 82.47 | 87.49 |
| BeatGAN | 90.13 | 88.94 | 87.97 | 92.04 | 87.67 | 89.75 | 96.06 | 70.20 | 81.07 | 96.06 | 70.20 | 81.07 | 89.15 | 67.81 | 76.63 | 83.30 |
| MAD-GAN | 88.51 | 90.45 | 88.03 | 85.96 | 88.38 | 86.98 | 70.47 | 78.41 | 74.23 | 79.18 | 54.23 | 63.85 | 96.40 | 54.74 | 69.82 | 76.58 |
| AnomalyTransformer | 87.26 | 89.70 | 88.46 | 92.31 | 93.29 | 92.80 | 86.34 | 91.08 | 88.64 | 80.57 | 82.34 | 81.45 | 85.36 | 88.03 | 86.67 | 87.60 |
| TFAD | 92.42 | 91.36 | 91.89 | 90.27 | 97.21 | 93.83 | 87.15 | 89.78 | 88.44 | 73.26 | 84.59 | 78.52 | 83.66 | 86.72 | 85.16 | 87.56 |
| TranAD | 89.06 | 89.82 | 87.85 | 95.06 | 91.47 | 92.20 | 89.51 | 92.97 | 91.15 | 70.25 | 72.66 | 68.86 | 82.24 | 85.02 | 83.60 | 84.73 |
| NPSR | 87.69 | 90.65 | 89.15 | 87.61 | 94.73 | 91.02 | 83.62 | 85.53 | 84.56 | 79.35 | 83.29 | 81.23 | 82.64 | 86.42 | 84.49 | 86.09 |
| DiffAD | 90.32 | 95.71 | 93.40 | 96.22 | 97.70 | 96.95 | 87.35 | 88.32 | 86.93 | 86.64 | 88.35 | 87.49 | 91.08 | 87.39 | 89.19 | 90.79 |
| ImDiffusion | 94.53 | 94.48 | 94.50 | 97.72 | 96.83 | 97.27 | 87.74 | 84.65 | 86.16 | 89.88 | 84.65 | 87.09 | 87.71 | 96.18 | 91.75 | 91.35 |
| D³R | 86.50 | 95.34 | 90.71 | 88.17 | 93.25 | 90.63 | 78.48 | 94.33 | 85.67 | 80.17 | 87.40 | 83.63 | 86.80 | 91.76 | 89.21 | 87.97 |
| Ours | 95.70 | 96.32 | 96.01 | 96.97 | 98.35 | 97.65 | 91.28 | 88.32 | 89.77 | 89.42 | 93.08 | 91.21 | 88.50 | 97.22 | 92.66 | 93.46 |

To ensure a more comprehensive comparison and provide a more convincing validation of MODEM's effectiveness, we also evaluate the detection performance using affiliation-based metrics (Wang et al., 2024) to avoid the illusion of progress caused by point adjustment (Kim et al., 2022). As shown in Tab. 6 of Appendix B.7, our MODEM achieves the highest F1-scores across three datasets with high non-stationarity, with an average improvement of 3.86% (from 80.34% to 84.20%). This suggests that the cross-resolution correlation enables a robust reconstruction signal in a coarse-to-fine manner, allowing most anomalous points to be effectively distinguished from normal non-stationary behavior.

## 4.4 ABLATION STUDIES

**Coarse-to-Fine Diffusion Model** We extended the diffusion model into a multi-resolution framework, incorporating cross-scale correlations into both the forward and reverse processes. The effects of various modifications are detailed in Tab. 3, where "w/o resolution", "w/o M-forward", and "w/o A-reverse" denote variants of MODEM that lack multi-resolution diffusion settings, modifications to the forward process, and adaptations to the reverse process, respectively. The data shows that removing the multi-resolution settings leads to a significant performance decrease across all datasets,

Table 2: Comparison of ADD (mean ± std.) performance across all five datasets. The best and second-best performances are highlighted in bold and underlined, respectively.

| Method | SMD | PSM | MSL | SMAP | SWaT | Average ADD |
|---|---|---|---|---|---|---|
| IForest | $90 \pm 1$ | $191 \pm 17$ | $123 \pm 28$ | $394 \pm 93$ | $539 \pm 20$ | $257 \pm 27$ |
| LSTM-AD | $87 \pm 1$ | $224 \pm 54$ | $115 \pm 29$ | $541 \pm 51$ | $627 \pm 4$ | $284 \pm 23$ |
| MSCRED | $32 \pm 0$ | $218 \pm 35$ | $109 \pm 30$ | $622 \pm 48$ | $1065 \pm 339$ | $365 \pm 76$ |
| GDN | $38 \pm 1$ | $148 \pm 0$ | $106 \pm 2$ | $402 \pm 4$ | $1478 \pm 0$ | $383 \pm 1$ |
| MTAD-GAT | $90 \pm 100$ | $182 \pm 0$ | $96 \pm 17$ | $542 \pm 2$ | $482 \pm 80$ | $256 \pm 33$ |
| OmniAnomaly | $26 \pm 1$ | $121 \pm 11$ | $93 \pm 2$ | $116 \pm 38$ | $550 \pm 48$ | $173 \pm 18$ |
| InterFusion | $\mathbf{22 \pm 2}$ | $40 \pm 10$ | $\mathbf{32 \pm 15}$ | $423 \pm 4$ | $454 \pm 141$ | $185 \pm 29$ |
| BeatGAN | $38 \pm 2$ | $166 \pm 11$ | $68 \pm 24$ | $345 \pm 23$ | $607 \pm 6$ | $226 \pm 13$ |
| MAD-GAN | $59 \pm 57$ | $122 \pm 2$ | $88 \pm 0$ | $404 \pm 20$ | $926 \pm 337$ | $293 \pm 69$ |
| TranAD | $\underline{24 \pm 0}$ | $127 \pm 4$ | $56 \pm 12$ | $291 \pm 2$ | $657 \pm 246$ | $210 \pm 46$ |
| DiffAD | $48 \pm 3$ | $134 \pm 7$ | $74 \pm 9$ | $368 \pm 22$ | $437 \pm 50$ | $212 \pm 18$ |
| ImDiffusion | $24 \pm 1$ | $\underline{28 \pm 1}$ | $\underline{46 \pm 4}$ | $\mathbf{98 \pm 31}$ | $\underline{350 \pm 43}$ | $\mathbf{104 \pm 14}$ |
| D³R | $65 \pm 16$ | $172 \pm 6$ | $90 \pm 12$ | $514 \pm 32$ | $643 \pm 46$ | $297 \pm 22$ |
| Ours | $27 \pm 2$ | $\mathbf{26 \pm 0}$ | $52 \pm 5$ | $\underline{102 \pm 8}$ | $\mathbf{343 \pm 26}$ | $\underline{109 \pm 9}$ |

with the average F1-score declining from 95.26% to 92.22%. This underscores the critical importance of integrating multi-resolution information for accurate anomaly detection. The guidance provided by lower-resolution data enables MODEM to better identify underlying temporal patterns and more effectively recover higher-resolution data. The alterations to the forward process and the adjustments to the reverse process contribute to an average performance improvement of 1.50% (from 93.76% to 95.26%) and 2.18% (from 93.08% to 95.26%), respectively. These outcomes confirm the efficacy of optimizing diffusion and sampling trajectories by leveraging correlations across different time scales.

**Frequency-Enhanced Decomposable Network** We conduct ablation experiments on three key components: frequency decomposition (w/o decomposition), the time-invariant encoder (w/o invariant), and the time-variant encoder (w/o variant), with results detailed in Tab. 3. The integration of frequency decomposition effectively disentangles time-varying dynamics in the frequency domain, enhancing our understanding of non-stationarity and resulting in an average F1-score improvement of 1.1% (from 94.16% to 95.26%). Furthermore, removing the time-invariant and time-variant encoders leads to a decline in performance across all datasets, with average F1-scores dropping to 94.40% and 93.10%, respectively. The time-invariant encoder captures globally shared information, while the time-variant encoder is designed to capture dynamic temporal patterns, working synergistically to achieve optimal performance. The more significant impact observed upon removing the time-variant encoder highlights its crucial role in Tab. 3: it accounts for an average improvement of 1.05% by adeptly learning complex periodic patterns across different receptive fields (see w/o DBTM). Additionally, the introduction of hierarchical transformers (w/o temporal and w/o spatial) within MODEM allows for the exploration of dependencies both temporally and across variables, leading to substantial improvements in F1-scores. Additional ablation studies on the number of DBTM and residual block are presented in Appendix B.6.

Table 3: Results of ablation studies on SMD, PSM, and SWaT datasets measured by F1-scores.

| Dataset | MODEM | w/o resolution | w/o M-forward | w/o A-reverse | w/o decomposition | w/o invariant | w/o variant | w/o DBTM | w/o temporal | w/o spatial |
|---|---|---|---|---|---|---|---|---|---|---|
| SMD | **96.01** | 93.02 | 94.88 | 93.84 | 94.73 | 94.96 | 93.15 | 95.01 | 93.33 | 94.69 |
| PSM | **97.65** | 95.20 | 96.67 | 96.32 | 96.90 | 97.09 | 95.77 | 96.96 | 95.47 | 95.68 |
| SWaT | **91.21** | 87.73 | 89.13 | 88.39 | 90.14 | 90.45 | 89.57 | 90.05 | 89.04 | 89.69 |
| Average | **94.96** | 91.98 | 93.56 | 92.85 | 93.92 | 94.16 | 92.83 | 94.01 | 92.61 | 93.35 |

## 4.5 EFFECTIVENESS ANALYSIS

**Coarse-to-Fine Diffusion Model** We model non-stationary time series across multiple resolutions ($R$) and have extended MODEM to the DDIM framework, which enables accelerated sampling through a sub-sequence of length $L$ at each resolution. As illustrated in Fig. 4, we explore the impact of varying $R$ and $L$ on the F1-scores in the SMD dataset. The results indicate that detection performance improves as $R$ increases from 1 to 4, highlighting that utilizing correlations across

different time scales enables the model to differentiate between anomalies and normal patterns more accurately. However, increasing $R$ beyond this point leads to a decline in performance, as some short-term anomalies are erroneously filtered out. Furthermore, as $L$ increases, MODEM's performance also improves, suggesting that a more comprehensive sampling process bolsters detection capabilities. The optimal configuration is found with $L = 20$ and $R = 4$, which not only maintains nearly intact performance but also achieves processing speeds comparable to previous methods (Chen et al., 2023). This balance between speed and accuracy renders MODEM highly effective for practical applications. Additional case studies and experimental results validating the effectiveness of multi-resolution modeling on the other four datasets can be found in Appendix B.8 and Appendix B.6, respectively.

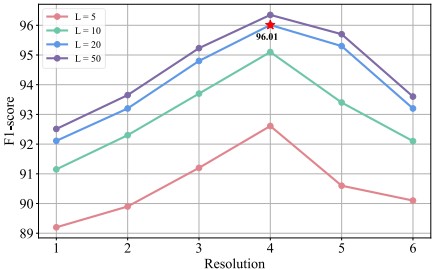
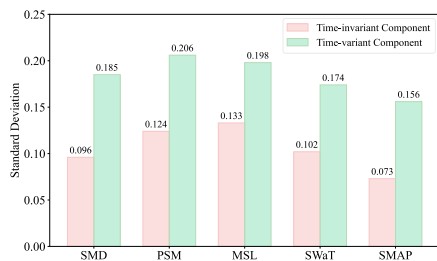

Figure 4: Results of various resolution and sampling steps on SMD dataset.

Figure 5: Results of standard deviation.

**Frequency Decomposition**  To evaluate the effectiveness of frequency decomposition, we sample 50 subsets from various periods in accordance with the methodology described in (Liu et al., 2024). We apply frequency decomposition to these subsets to extract the time-invariant and time-variant components. Separate linear regressions are conducted on each component, and the standard deviation of the regression weights is utilized to assess the interdependencies between the components. The results, illustrated in Fig. 5, show that the standard deviations for the time-variant components are consistently larger than those for the time-invariant components across all datasets. This demonstrates that frequency decomposition effectively separates the two types of components, confirming its utility in enhancing time series analysis. More experimental results and clarifications on our frequency-enhanced decomposable network are included in Appendix B.8.

### 4.6 GENERALIZATION PERFORMANCE

To assess the generalization ability of MODEM, we conduct forecasting and imputation experiments across multiple non-stationary datasets. We apply MODEM in an unconditional manner to non-stationary time series forecasting and imputation following (Kollovieh et al., 2024), and report the average *continuous ranked probability scores* (CRPS) (Gneiting & Raftery, 2007) across three independent runs in Tab. 15 and Tab. 16 of Appendix B.11, as well as the detailed discussion on experimental setup can be found in Appendix B). The results demonstrate that MODEM performs competitively against SOTA methods, despite not being specifically designed for forecasting tasks. We also extend the proposed multi-resolution diffusion paradigm to CSDI (Tashiro et al., 2021) and TSDiff (Kollovieh et al., 2024) (denoted as CSDI-MR and TSDiff-MR). The improved forecasting and imputation performance validate that correlations across different time scales facilitate the model to better capture diverse temporal patterns.

## 5 CONCLUSION

This paper introduces a novel "Multi-Resolution Decomposable Diffusion Model" that delves deeply into non-stationary time series to enhance anomaly detection performance. Our method innovatively incorporates multi-resolution correlation information into a Coarse-to-Fine Diffusion Model to optimize the diffusion trajectories, effectively capturing the non-stationarity to better differentiate between anomalies and normal patterns. Additionally, we design a frequency-enhanced decomposable network, which separates time-invariant and time-variant components in the frequency domain. Extensive experiments conducted on five real-world datasets demonstrate that MODEM significantly outperforms state-of-the-art unsupervised anomaly detection methods.

## 6 ACKNOWLEDGEMENTS

This work was supported by the National Natural Science Foundation of China (No. 62272157, No. U21A20518, No. U23A20341).

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

# A MATHEMATICAL DERIVATIONS

## A.1 DERIVATION OF MULTI-RESOLUTION DIFFUSION TRANSITIONS

Given the diffusion trajectory modified by cross-scale correlation, which is defined as Eq. (1), we now derive the multi-resolution diffusion transitions:

$$q_\phi(\mathbf{x}_k^r|\mathbf{x}_{k-1}^r, \mathbf{x}_0^r, \mathbf{x}_0^{r+1}) = \mathcal{N}(\sqrt{\alpha_k}\mathbf{x}_{k-1}^r + \gamma_k\mathcal{E}_\phi(\mathbf{x}_0^r, \mathbf{x}_0^{r+1}, k) - \sqrt{\alpha_k}\gamma_{k-1}\mathcal{E}_\phi(\mathbf{x}_0^r, \mathbf{x}_0^{r+1}, k-1), \beta_k\mathbf{I}). \tag{9}$$

We can achieve this by proving that Lemma. 1, which has been proposed in (Zhang et al., 2023; Yang et al., 2024).

**lemma 1** *Given the forward process defined as* $q(\mathbf{x}_1^r, \mathbf{x}_2^r, ..., \mathbf{x}_k^r|\mathbf{x}_0^r) = \prod_{t=1}^T q(\mathbf{x}_k^r|\mathbf{x}_{k-1}^r, \mathbf{x}_0^r, \mathbf{x}_0^{r+1})$, *where the diffusion transitions* $q(\mathbf{x}_k^r|\mathbf{x}_{k-1}^r, \mathbf{x}_0^r, \mathbf{x}_0^{r+1})$ *are defined as:*

$$q_\phi(\mathbf{x}_k^r|\mathbf{x}_0^r, \mathbf{x}_0^{r+1}) = \mathcal{N}(\mathbf{x}_k^r, \sqrt{\bar{\alpha}_k}\mathbf{x}_0^r + \gamma_k\mathcal{E}_\phi(\mathbf{x}_0^r, \mathbf{x}_0^{r+1}, k), (1-\bar{\alpha}_k)\mathbf{I}). \tag{10}$$

**Proof 1** *Inspired by previous work (Yang et al., 2024), we can prove this lemma by induction. Assume that at time* $k$, *both* $q(\mathbf{x}_k^r|\mathbf{x}_{k-1}^r, \mathbf{x}_0^r, \mathbf{x}_0^{r+1})$ *and* $q(\mathbf{x}_{k-1}^r|\mathbf{x}_0^r, \mathbf{x}_0^{r+1})$ *adhere to their respective distributions as in Eq.(3) and Eq. (1). We need to prove that* $q(\mathbf{x}_k^r|\mathbf{x}_0^r, \mathbf{x}_0^{r+1}) = \mathcal{N}(\mathbf{x}_k^r; \sqrt{\bar{\alpha}_k}\mathbf{x}_0^r + \gamma_k\mathcal{E}_\phi(\mathbf{x}_0^r, \mathbf{x}_0^{r+1}, k), (1-\bar{\alpha}_k)\boldsymbol{I})$.

*We can rewrite* $q(\mathbf{x}_k^r|\mathbf{x}_{k-1}^r, \mathbf{x}_0^r, \mathbf{x}_0^{r+1})$ *and* $q(\mathbf{x}_{k-1}^r|\mathbf{x}_0^r, \mathbf{x}_0^{r+1})$ *as follows:*

$$\mathbf{x}_k^r = \sqrt{\alpha_k}\mathbf{x}_{k-1}^r + \gamma_k\mathcal{E}_\phi(\mathbf{x}_0^r, \mathbf{x}_0^{r+1}, k) - \sqrt{\alpha_k}\gamma_{k-1}\mathcal{E}_\phi(\mathbf{x}_0^r, \mathbf{x}_0^{r+1}, k-1) + \sqrt{\beta_k}\epsilon_1, \tag{11}$$

$$\mathbf{x}_{k-1}^r = \sqrt{\bar{\alpha}_{t-1}}\mathbf{x}_0^r + \gamma_{k-1}\mathcal{E}_\phi(\mathbf{x}_0^r, \mathbf{x}_0^{r+1}, k-1) + \sqrt{1-\bar{\alpha}_{t-1}}\epsilon_2, \tag{12}$$

*where* $\epsilon_1$ *and* $\epsilon_2$ *denote independent standard Gaussian variables. Substituting* $\mathbf{x}_{k-1}^r$ *from the latter equation into the former, we obtain:*

$$\begin{aligned}\mathbf{x}_k^r &= \sqrt{\bar{\alpha}_k}\mathbf{x}_0^r + \gamma_k\mathcal{E}_\phi(\mathbf{x}_0^r, \mathbf{x}_0^{r+1}, k) + \sqrt{\beta_k}\epsilon_1 + \sqrt{\alpha_k*(1-\bar{\alpha}_{k-1})}*\epsilon_2 \\ &= \sqrt{\bar{\alpha}_k}\mathbf{x}_0^r + \gamma_k\mathcal{E}_\phi(\mathbf{x}_0^r, \mathbf{x}_0^{r+1}, k) + \sqrt{\alpha_t(1-\bar{\alpha}k-1)+\beta_k}\epsilon, \end{aligned} \tag{13}$$

*where* $\epsilon$ *is a Gaussian noise resulting from a linear combination of* $\epsilon_1$ *and* $\epsilon_2$. *To this end,* $q_\phi(\mathbf{x}_k^r|\mathbf{x}_{k-1}^r, \mathbf{x}_0^r, \mathbf{x}_0^{r+1})$ *with mean* $\sqrt{\bar{\alpha}_k}\mathbf{x}_0^r + \gamma_k\mathcal{E}_\phi(\mathbf{x}_0^r, \mathbf{x}_0^{r+1}, k)$ *and variance* $\epsilon$ *admits the expected distribution.*

## A.2 DERIVATION OF POSTERIOR DISTRIBUTIONS OF MULTI-RESOLUTION DIFFUSION PROCESS

Given the modified diffusion trajectories in Eq. (1) and the diffusion transitions in Eq. (2), we now derive the posterior distributions of multi-resolution diffusion process:

$$\begin{aligned}q_\phi(\mathbf{x}_{k-1}^r|\mathbf{x}_k^r, \mathbf{x}_0^r, \mathbf{x}_0^{r+1}) =& \mathcal{N}(\frac{\sqrt{\bar{\alpha}_{k-1}}\beta_k}{1-\bar{\alpha}_k}\mathbf{x}_0^r + \frac{\sqrt{\alpha_k}(1-\bar{\alpha}_{k-1})}{1-\bar{\alpha}_k}(\mathbf{x}_k^r - \gamma_k\mathcal{E}_\phi(\mathbf{x}_0^r, \mathbf{x}_0^{r+1}, k)) \\ &+ \gamma_{k-1}\mathcal{E}_\phi(\mathbf{x}_0^r, \mathbf{x}_0^{r+1}, k-1), \frac{(1-\bar{\alpha}_{k-1})\beta_k}{1-\bar{\alpha}_k}\mathbf{I}).\end{aligned} \tag{14}$$

**Proof 2** *By Bayes's rule (Ho et al., 2020), we have:*

$$q(\mathbf{x}_{k-1}^r|\mathbf{x}_k^r, \mathbf{x}_0^r, \mathbf{x}_0^{r+1}) = \frac{q(\mathbf{x}_{k-1}^r|\mathbf{x}_0^r, \mathbf{x}_0^{r+1})q(\mathbf{x}_k^r|\mathbf{x}_{k-1}^r, \mathbf{x}_0^r, \mathbf{x}_0^{r+1})}{q(\mathbf{x}_k^r|\mathbf{x}_0^r, \mathbf{x}_0^{r+1})}. \tag{15}$$

*Given that the numerator and denominator are both Gaussian, the posterior distribution is also Gaussian (Song et al., 2020), and we can proceed to calculate its mean and variance:*

$$\begin{aligned}q(\mathbf{x}_{k-1}^r|\mathbf{x}_k^r, \mathbf{x}_0^r, \mathbf{x}_0^r) =& \frac{\mathcal{N}(\mathbf{x}_{k-1}^r, \sqrt{\bar{\alpha}_{k-1}}\mathbf{x}_0^r + \gamma_k\mathcal{E}_\phi(\mathbf{x}_0^r, \mathbf{x}_0^{r+1}, k), (1-\bar{\alpha}_{t-1})\boldsymbol{I})}{\mathcal{N}(\mathbf{x}_k^r, \sqrt{\bar{\alpha}_k}\mathbf{x}_0^r + \gamma_{k-1}\mathcal{E}_\phi(\mathbf{x}_0^r, \mathbf{x}_0^{r+1}, k-1), (1-\bar{\alpha}_{t-1})\boldsymbol{I})}* \\ &\frac{\mathcal{N}(\boldsymbol{x}_t, \sqrt{\alpha_t}\boldsymbol{x}_{t-1} + \gamma_k\mathcal{E}_\phi(\mathbf{x}_0^r, \mathbf{x}_0^{r+1}, k) - \sqrt{\alpha_t}\gamma_{k-1}\mathcal{E}_\phi(\mathbf{x}_0^r, \mathbf{x}_0^{r+1}, k-1), \beta_t\boldsymbol{I})}{\mathcal{N}(\mathbf{x}_k^r, \sqrt{\bar{\alpha}_k}\mathbf{x}_0^r + \gamma_{k-1}\mathcal{E}_\phi(\mathbf{x}_0^r, \mathbf{x}_0^{r+1}, k-1), (1-\bar{\alpha}_{t-1})\boldsymbol{I})},\end{aligned} \tag{16}$$

*Dropping the constants that are unrelated to $\mathbf{x}_0^r$, $\mathbf{x}_k^r$, $\mathbf{x}_{k-1}^r$, and $\mathbf{x}_0^{r+1}$, we have:*

$$
\begin{aligned}
q(\mathbf{x}_{k-1}^r|\mathbf{x}_k^r,\mathbf{x}_0^r,\mathbf{x}_0^{r+1}) \propto \exp\{ &-\frac{(\mathbf{x}_{k-1}^r - \sqrt{\bar{\alpha}_{k-1}}\mathbf{x}_0^r - \gamma_k\mathcal{E}_\phi(\mathbf{x}_0^r,\mathbf{x}_0^{r+1},k))^2}{2(1-\bar{\alpha}_{k-1})} \\
&+ \frac{(\mathbf{x}_k^r - \sqrt{\bar{\alpha}_k}\mathbf{x}_0^r - \gamma_k\mathcal{E}_\phi(\mathbf{x}_0^r,\mathbf{x}_0^{r+1},k))^2}{2(1-\bar{\alpha}_k)} \\
&- \frac{(\mathbf{x}_k^r - \sqrt{\alpha_k}\mathbf{x}_{k-1}^r - \gamma_k\mathcal{E}_\phi(\mathbf{x}_0^r,\mathbf{x}_0^{r+1},k) + \sqrt{\alpha_k}\gamma_k\mathcal{E}_\phi(\mathbf{x}_0^r,\mathbf{x}_0^{r+1},k))^2}{2\beta_k}\} \\
= \exp\{C(k,r) &- \frac{1}{2}(\frac{1}{1-\bar{\alpha}_{k-1}} + \frac{\alpha_k}{\beta_k}) * {\mathbf{x}_{k-1}^r}^2 + \mathbf{x}_{k-1}* \\
[\frac{(\sqrt{\bar{\alpha}_{k-1}}\mathbf{x}_0^r + \gamma_k\mathcal{E}_\phi(\mathbf{x}_0^r,\mathbf{x}_0^{r+1},k))}{1-\bar{\alpha}_{k-1}} &+ \sqrt{\alpha_k}\frac{(\mathbf{x}_k - \gamma_k\mathcal{E}_\phi(\mathbf{x}_0^r,\mathbf{x}_0^{r+1},k) + \sqrt{\alpha_k}\gamma_{k-1}\mathcal{E}_\phi(\mathbf{x}_0^r,\mathbf{x}_0^{r+1},k-1))}{\beta_k}]\} \\
= \exp\{C(k,r) &- \frac{1}{2}(\frac{1}{1-\bar{\alpha}_{k-1}} + \frac{\alpha_k}{\beta_k}) * {\mathbf{x}_{k-1}^r}^2 + \mathbf{x}_{k-1}* \\
[\frac{(\sqrt{\bar{\alpha}_{k-1}}}{1-\bar{\alpha}_{k-1}}\mathbf{x}_0^r + \frac{\sqrt{\alpha_k}}{\beta_k}&(\mathbf{x}_k^r - \gamma_k\mathcal{E}_\phi(\mathbf{x}_0^r,\mathbf{x}_0^{r+1},k)) + (\frac{1}{1-\bar{\alpha}_{k-1}} + \frac{\alpha_k}{\beta_k}) * \gamma_{k-1}\mathcal{E}_\phi(\mathbf{x}_0^r,\mathbf{x}_0^{r+1},k-1)]\},
\end{aligned}
\tag{17}
$$

*where $C(k,r)$ is a constant term with respect to $\mathbf{x}_{t-1}^r$. With some algebraic derivation (Yang et al., 2024), this can be simplified to:*

$$
\begin{aligned}
q_\phi(\mathbf{x}_{k-1}^r|\mathbf{x}_k^r,\mathbf{x}_0^r,\mathbf{x}_0^{r+1}) = \mathcal{N}(&\frac{\sqrt{\bar{\alpha}_{k-1}}\beta_k}{1-\bar{\alpha}_k}\mathbf{x}_0^r + \frac{\sqrt{\alpha_k}(1-\bar{\alpha}_{k-1})}{1-\bar{\alpha}_k}(\mathbf{x}_k^r - \gamma_k\mathcal{E}_\phi(\mathbf{x}_0^r,\mathbf{x}_0^{r+1},k)) \\
&+ \gamma_{k-1}\mathcal{E}_\phi(\mathbf{x}_0^r,\mathbf{x}_0^{r+1},k-1), \frac{(1-\bar{\alpha}_{k-1})\beta_k}{1-\bar{\alpha}_k}\mathbf{I}).
\end{aligned}
\tag{18}
$$

### A.3 DERIVATION OF TRAINING OBJECTIVE

According to Eq. (1), $\mathbf{x}_0^r$ can be rewritten as:

$$
\mathbf{x}_0^r = \frac{1}{\sqrt{\bar{\alpha}_k}}(\mathbf{x}_k^r - \gamma_k\mathcal{E}_\phi(\mathbf{x}_0^r,\mathbf{x}_0^{r+1},k) - \sqrt{1-\bar{\alpha}_k}\epsilon),
\tag{19}
$$

where $\epsilon$ denotes gaussian noise. Then we can obatin:

$$
\epsilon = \frac{\mathbf{x}_k^r - \sqrt{\bar{\alpha}_k}\mathbf{x}_0^r}{\sqrt{1-\bar{\alpha}_k}} - \frac{\gamma_k\mathcal{E}_\phi(\mathbf{x}_0^r,\mathbf{x}_0^{r+1},k)}{\sqrt{1-\bar{\alpha}_k}}.
\tag{20}
$$

Since the second term is available, we employ a denoising network $f_\theta(\mathbf{x}_k^r,k,r)$ to predict the first term for training. Then we can obtain the predicted posterior distributions:

$$
\begin{aligned}
p_\theta(\mathbf{x}_{k-1}^r|\mathbf{x}_k^r,\mathbf{x}_0^{r+1}) = \mathcal{N}(&\frac{1}{\sqrt{\alpha_k}}[\mathbf{x}_k^r - \frac{\beta_k}{\sqrt{1-\bar{\alpha}_k}}f_\theta(\mathbf{x}_k^r,k,r)] - \frac{\sqrt{\alpha_k(1-\bar{\alpha}_{k-1})}}{1-\bar{\alpha}_k}\gamma_k\mathcal{E}_\phi(\mathbf{x}_0^r,\mathbf{x}_0^{r+1},k) \\
&+ \gamma_{k-1}\mathcal{E}_\phi(\mathbf{x}_0^r,\mathbf{x}_0^{r+1},k-1), \frac{(1-\bar{\alpha}_{k-1})\beta_k}{1-\bar{\alpha}_k}\mathbf{I}),
\end{aligned}
\tag{21}
$$

Revisiting the objective of diffusion model (Song et al., 2020), we insteadly minimize the KL-divergence (Van Erven & Harremos, 2014) $D_{KL}(q_\phi(\mathbf{x}_{k-1}^r|\mathbf{x}_k^r,\mathbf{x}_0^r,\mathbf{x}_0^{r+1})\|p_\theta(\mathbf{x}_{k-1}^r|\mathbf{x}_k^r,\mathbf{x}_0^{r+1}))$.

With Eq. (3) and Eq. (4), we can obtain the training objective:

$$
\begin{aligned}
\mathcal{L}_{\theta,\phi,k,r} &= D_{KL}(q_\phi(\mathbf{x}_{k-1}^r|\mathbf{x}_k^r,\mathbf{x}_0^r,\mathbf{x}_0^{r+1})\|p_\theta(\mathbf{x}_{k-1}^r|\mathbf{x}_k^r,\mathbf{x}_0^{r+1})) \\
&= \sum_{k=1}^K \eta_k\mathbb{E}_{\mathbf{x}_0^r,\epsilon,r}\big[\|\epsilon_\theta(\mathbf{x}_k^r,k,r) - \frac{\mathbf{x}_k^r - \sqrt{\bar{\alpha}_k}\mathbf{x}_0^r}{\sqrt{1-\bar{\alpha}_k}}\|^2\big],
\end{aligned}
\tag{22}
$$

where $\mathbf{x}_k^r = \sqrt{\bar{\alpha}_k}\mathbf{x}_0^r + \gamma_k\mathcal{E}_\phi(\mathbf{x}_0^r,\mathbf{x}_0^{r+1},k) + \sqrt{1-\bar{\alpha}_k}\epsilon$ and $\eta_k = \frac{\beta_k}{2\alpha_k(1-\bar{\alpha}_{k-1})}$ is a loss weight.

## A.4 Derivation of Accelerated Sampling

Given $\mathbf{x}_k^r$, we can obtain $\mathbf{x}_{k-1}^r$ via:

$$
\begin{aligned}
\mathbf{x}_{k-1}^r =& \sqrt{\bar{\alpha}_{k-1}}\hat{\mathbf{x}}_0^r + \sqrt{1 - \bar{\alpha}_{k-1} - \sigma_k^2} \cdot \frac{\mathbf{x}_k^r - \sqrt{\bar{\alpha}_k}\hat{\mathbf{x}}_0^r}{\sqrt{1 - \bar{\alpha}_k}} \\
&- \gamma_k \mathcal{E}_\phi(\hat{\mathbf{x}}_0^r, \hat{\mathbf{x}}_0^{r+1}, k) \cdot \frac{1 - \bar{\alpha}_{k-1} - \sigma_k^2}{\sqrt{1 - \bar{\alpha}_k}} + \gamma_{k-1}\mathcal{E}_\phi(\hat{\mathbf{x}}_0^r, \hat{\mathbf{x}}_0^{r+1}, k-1).
\end{aligned} \tag{23}
$$

**Proof 3** *We can prove this by induction following (Zhang et al., 2023). Assume that at time $k$, the posterior and marginal distributions admit the expected distributions, then we need to prove that at time $k-1$, $q_\phi(\mathbf{x}_{k-1}^r|\mathbf{x}_0^r, \mathbf{x}_0^{r+1})$ also has the expected distribution. We can rewrite the posterior and marginal distribution:*

$$
\begin{aligned}
\mathbf{x}_{k-1}^r =& \sqrt{\bar{\alpha}_{k-1}}\mathbf{x}_0^r + \sqrt{1 - \bar{\alpha}_{k-1} - \sigma_k^2} * \frac{\mathbf{x}_k^r - \sqrt{\bar{\alpha}_k}\mathbf{x}_0^r}{\sqrt{1 - \bar{\alpha}_k}} \\
&- \gamma_k \mathcal{E}_\phi(\mathbf{x}_0^r, \mathbf{x}_0^{r+1}, k) * \frac{\sqrt{1 - \bar{\alpha}_{k-1} - \sigma_k^2}}{\sqrt{1 - \bar{\alpha}_k}} + \gamma_{k-1}\mathcal{E}_\phi(\mathbf{x}_0^r, \mathbf{x}_0^{r+1}, k-1) + \sigma_k \epsilon_1,
\end{aligned} \tag{24}
$$

$$
\mathbf{x}_k^r = \sqrt{\bar{\alpha}_k}\mathbf{x}_0^r + \gamma_k \mathcal{E}_\phi(\mathbf{x}_0^r, \mathbf{x}_0^{r+1}, k) + \sqrt{1 - \bar{\alpha}_k}\epsilon_2, \tag{25}
$$

*where $\epsilon_1, \epsilon_2$ are standard gaussian noises. Plugging in $\mathbf{x}_k^r$, we have:*

$$
\begin{aligned}
\mathbf{x}_{k-1}^r =& \sqrt{\bar{\alpha}_{k-1}}\mathbf{x}_0^r \\
&+ \gamma_k \mathcal{E}_\phi(\mathbf{x}_0^r, \mathbf{x}_0^{r+1}, k) * \frac{\sqrt{1 - \bar{\alpha}_{k-1} - \sigma_k^2}}{\sqrt{1 - \bar{\alpha}_k}} - \gamma_k \mathcal{E}_\phi(\mathbf{x}_0^r, \mathbf{x}_0^{r+1}, k) * \frac{\sqrt{1 - \bar{\alpha}_{k-1} - \sigma_k^2}}{\sqrt{1 - \bar{\alpha}_k}} \\
&+ \gamma_{k-1}\mathcal{E}_\phi(\mathbf{x}_0^r, \mathbf{x}_0^{r+1}, k-1) + \sigma_k \epsilon_1 + \sqrt{1 - \bar{\alpha}_{k-1} - \sigma_k^2}\epsilon_2 \\
=& \sqrt{\bar{\alpha}_{k-1}}\mathbf{x}_0^r + \gamma_{k-1}\mathcal{E}_\phi(\mathbf{x}_0^r, \mathbf{x}_0^{r+1}, k-1) + \sigma_k \epsilon_1 + \sqrt{1 - \bar{\alpha}_{k-1} - \sigma_k^2}\epsilon_2.
\end{aligned} \tag{26}
$$

*Since the variance of $(\sigma_t \epsilon_1 + \sqrt{1 - \bar{\alpha}_{t-1} - \sigma_t^2}\epsilon_2)^2 = (1 - \bar{\alpha}_{t-1})\mathbf{I}$, we have the expected sampling.*

## B Experimental Details

### B.1 Datasets

We use the following five datasets for anomaly detection experiments:

- **SMD (Server Machine Dataset)** (Su et al., 2019): The SMD dataset is collected from a large internet company and includes 5 weeks of data from 28 server machines with 38 sensors each. The initial 5 days consist solely of normal data, while anomalies are intermittently introduced over the last 5 days.

- **PSM (Pooled Server Metrics)** (Abdulaal et al., 2021): The PSM dataset is collected internally from multiple application server nodes at eBay. It consists of 13 weeks of training data and 8 weeks of testing data.

- **MSL (Mars Science Laboratory)** (Hundman et al., 2018) and **SMAP (Soil Moisture Active Passive satellite)** (Entekhabi et al., 2010): The MSL and SMAP datasets are publicly available datasets collected by NASA. They contain telemetry anomaly data derived from the Incident Surprise Anomaly (ISA) reports of spacecraft monitoring systems. The MSL dataset has 55 dimensions, while the SMAP dataset has 25 dimensions. The training sets for both datasets include unlabeled anomalies.

- **SWaT (Secure Water Treatment)** (Mathur & Tippenhauer, 2016): The SWaT dataset is collected over 11 days from a scaled-down water treatment testbed with 51 sensors. For the first 7 days, only normal data were generated. During the last 4 days, 41 anomalies were injected using various attack methods.

Table 4: Datasets used for anomaly detection experiments.

| Dataset | Entities | Dimensions | Train # | Test # | Anomaly Rate (%) |
|---------|----------|------------|---------|--------|------------------|
| SMD | 28 | 38 | 708405 | 708420 | 4.16 |
| PSM | 1 | 25 | 132481 | 87841 | 27.76 |
| MSL | 27 | 55 | 58317 | 73729 | 10.48 |
| SMAP | 55 | 25 | 140825 | 444035 | 12.83 |
| SWaT | 1 | 51 | 495000 | 449919 | 12.14 |

We present the statistics of these datasets in Table. 4. Train # and Test # denote the number of training and testing data, respectively. Anomaly Rate is the ratio between the sum of all anomaly points and sum of all test points.

We use the following three non-stationary datasets for forecasting and imputation experiments to assess the generalization ability of MODEM.

- **Traffic** (Tashiro et al., 2021): The Traffic dataset records the hourly road occupancy rates generated by sensors in the San Francisco Bay area freeways.
- **Exchange** (Shen & Kwok, 2023): The Exchange dataset describes the daily exchange rates of eight countries (Australia, British, Canada, Switzerland, China, Japan, New Zealand, and Singapore).
- **KDDCup** (Kollovieh et al., 2024): The KDDCup is a dataset of the air quality indices (AQIs) of Beijing and London used in the KDD Cup 2018.

For imputation performance evaluation, we examine three scenarios following (Kollovieh et al., 2024): (1) random missing, where values are missing sporadically, (2) blackout missing at the beginning of the context window, involving a sequence of consecutive missing values, and (3) blackout missing at the end of the context window. We report the average performance of three conditions.

## B.2 BASELINES

We introduce the following state-of-the-art time series anomaly detection methods for extensive comparisons:

- **Isolation Forest** (Li et al., 2022a): constructs 3D features (text, reviewer behavior, deceptive ratings) and integrates feature selection to detect fake reviews.
- **LSTM-AD** (Malhotra et al., 2015): possesses long-term memory capabilities, and for the first time, hierarchical recurrent processing layers have been combined to detect anomalies in univariate time series without using labels for training.
- **MSCRED** (Zhang et al., 2019): designs an attention-based ConvLSTM network to capture temporal trends, and a convolutional autoencoder is used to encode and reconstruct the signature matrix instead of relying on the time series explicitly.
- **OmniAnomaly** (Su et al., 2019): is a Variational Autoencoder that performs anomaly detection by computing the reconstruction probability and quantifying interpretability based on the reconstruction probability of each feature.
- **InterFusion** (Li et al., 2021): uses a hierarchical variational autoencoder with two random latent variables to learn metrics and temporal representations and by relying on a "reconstruction input" to compress the MTS.
- **GDN** (Deng & Hooi, 2021): utilizes the nodes and edges of the GNN to capture sensor features and spatial information, respectively. It then leverages this data to predict sensor behavior based on the attention function of adjacent sensors.
- **MST-GAT** (Ding et al., 2023): uses a multimodal graph attention network and a temporal convolutional network to capture spatiotemporal correlations in multimodal time series.
- **BeatGAN** (Zhou et al., 2019): uses a group of autoencoders and GANs in cases where tags are not available, which accurately detect anomalies in both ECG and sensor data.

- **MAD-GAN** (Li et al., 2019): uses LSTM-RNN as a generator and discriminator to capture temporal relationships in Gans, while using reconstruction and discriminant losses to detect anomalies.

- **Anomaly Transformer** (Xu et al., 2022): captures association differences by modeling prior and sequential associations for each timestamp, making rare exceptions easier to distinguish.

- **TFAD** (Zhang et al., 2022): introduces time series decomposition and data augmentation mechanisms into the designed time-frequency architecture, enhancing both performance and interpretability.

- **TranAD** (Tuli et al., 2022): adopts a two-step reconstruction method, introduces the attention mechanism into Transformer model, and integrates adversarial training.

- **NPSR** (Lai et al., 2024): proposes a framework for unsupervised time series anomaly detection using point-based and sequence-based reconstruction models.

- **DiffAD** (Xiao et al., 2023): designs a novel denoising diffusion-based imputation method to improve the imputation performance of missing values with conditional weight-incremental diffusion.

- **ImDiffusion** (Chen et al., 2023): combines time series imputation and diffusion models to achieve accurate and robust anomaly detection.

- **$D^3R$** (Wang et al., 2024): tackles the drift via decomposition and reconstruction, overcoming the limitation of the local sliding window.

We introduce the following state-of-the-art time series forecasting and imputation methods for comparison to assess the generalization ability of MODEM:

- **CSDI** (Tashiro et al., 2021): is a self-supervised method that uses the observed value as conditional information to imputation the masked time series.

- **TSDiff** (Kollovieh et al., 2024): is an unconditionally trained diffusion model for time series and a mechanism to condition TSDiff during inference for arbitrary forecasting tasks (observation self-guidance).

To ensure fair comparisons, we first reference the best-reported values from the original papers or other publications. If these values are unavailable, we report the results reproduced on our machine using the publicly available codes. Moreover, we assume that the thresholds of these SOTA methods have been optimized for optimal performance.

Notably, TSDiff is developed for univariate time series generation, and the selected datasets—Traffic, Exchange, and KDDCup—are widely used, preprocessed univariate datasets derived from traffic, financial exchange rates, and climate domains, respectively. These datasets, after processing, can be obtained from Appendix B.1 of the TSDiff paper. When extending the proposed multi-resolution approach to TSDiff, we preserved its original network structure and hyperparameter settings as presented in the original paper and code. The only additions were the resolution scale $R$ and an extra cross-attention layer to capture correlations across different resolutions. In practical implementation, we modified the original diffusion and sampling processes so that TSDiff first generates low-resolution data and then refines it to high-resolution data.

Moreover, similar to TSDiff, our proposed MODEM is also an unconditional diffusion model. When applying MODEM to univariate datasets, the Spatial Transformer components in both the time-invariant and time-variant encoders were omitted, and the model's input dimensions were adjusted based on the specific context length and prediction length of each dataset. The resolution scale, context length, and prediction length for the Traffic, Exchange, and KDDCup datasets are set to $(3, 336, 24)$, $(3, 360, 30)$, and $(4, 312, 48)$, respectively. These modifications ensure seamless deployment of MODEM on univariate data.

### B.3 EVALUATION METRICS

We use the following metrics for evaluating the performance of time series anomaly detection methods:

- **Precision**: It measures the proportion of correctly detected anomalies among all time points flagged as anomalies:

$$\text{Precision} = \frac{\text{TP}}{\text{TP} + \text{FP}}, \tag{27}$$

  where TP denotes the number of true anomalies correctly detected, and FP denotes the number of normal time points incorrectly identified as anomalies.

- **Recall**: This metric is also named as sensitivity or true positive rate. It measures the proportion of actual anomalies that are correctly detected by the algorithm:

$$\text{Recall} = \frac{\text{TP}}{\text{TP} + \text{FN}}, \tag{28}$$

  where FN denotes the number of true anomalies that are not detected.

- **F1-score**: F1-score is the harmonic mean of **Precision** and **Recall**, which can balance both false positives and false negatives:

$$\text{F1-score} = 2 \times \frac{\text{Precision} \times \text{Recall}}{\text{Precision} + \text{Recall}}. \tag{29}$$

- **Average Sequence Detection Delay (ADD)** (Tuli et al., 2022): It is used for evaluating the speed and timeliness of time series anomaly detection algorithm:

$$\text{ADD} = \frac{1}{S} \sum_{i=1}^{S} (\mathcal{T}_i - \rho_i), \tag{30}$$

  where $\rho_i$ denotes the initial time of anomalous span $i$, $\mathcal{T}_i \geq \rho_i$ denotes the corresponding detection delay time by the anomaly detection algorithm. $S$ is the total number of anomalous spans. A small ADD signifies a more timely detection of anomalies.

## B.4 ENSEMBLE INFERENCE

Considering the multi-resolution setting, we extend the ensemble inference (Chen et al., 2023) to combine the reconstruction errors $\{E_{1,1}, ..., E_{t,1}, ..., E_{t,r}, ..., E_{T,R}\}$ at different diffusion steps $t$ and resolution scales $r$ for enhanced anomaly detection. First, the anomaly label $Y_{t,r}$ at step $t, r$ is computed based on the reconstruction error:

$$Y_{t,r} = \mathbb{1}(E_{t,r} \geq \pi_{t,r}), \text{ where } \pi_{t,r} = \frac{\Sigma E_{T,R}}{\Sigma E_{t,r}} \cdot \pi_{T,R}, \tag{31}$$

where $\pi_{T,R}$ denotes the upper percentage of reconstruction errors at final step $T, R$. This approach allows us to assess the reconstruction quality at each step relative to the final reconstruction error. If the ratio $\frac{\Sigma E_{T,R}}{\Sigma E_{t,r}}$ is small, it indicates poor reconstruction quality, and the upper percentage of anomaly labels decreases accordingly, retaining only the time points with high reconstruction errors and confidence. After obtaining the prediction labels $Y_{t,r} = \{y_{t,r}^1, y_{t,r}^2, ..., y_{t,r}^l\}$ (here $y_{t,r}^l = 1$ denotes that the data at time point $l$ is predicted as an anomaly, and $y_{t,r}^l = 0$ otherwise), we use a voting scheme to aggregate them. If the number of anomaly votes $\mathbf{v}_l = \sum_{t=1}^{T} \sum_{r=1}^{R} y_{t,r}^l$ at time point $l$ exceeds the voting threshold $\zeta$, that is:

$$y_l = \mathbb{1}(\mathbf{v}_l > \zeta), \tag{32}$$

it will be classified as an anomaly data. Since the reconstruction errors in the earlier diffusion steps are typically larger, we only aggregate the denoising results from the last 10 diffusion steps for each resolution.

It is well known that there is an inherent trade-off between precision and recall. By setting a higher voting threshold $\zeta$, the model reduces false positives (i.e., higher precision) but risks missing some actual anomalies (i.e., lower recall). Conversely, a lower threshold allows the model to detect more anomalies (i.e., higher recall) but may result in more false positives (i.e., lower precision).

B.5 NETWORK STRUCTURE

The detailed architecture of Time-Invariant Encoder is presented in Figure. 6, it comprises multiple residual blocks. The residual block first uses a $1 \times 1$ convolution kernel to encode the time-invariant components obtained from frequency decomposition. The diffusion step $k$ and resolution scale $r$ are fed into simple MLPs to obtain corresponding embeddings, which are added to the convolution result as supplementary information. Subsequently, it employs hierarchical transformers to further explore the intra-series temporal features and inter-series dependencies among various variables.

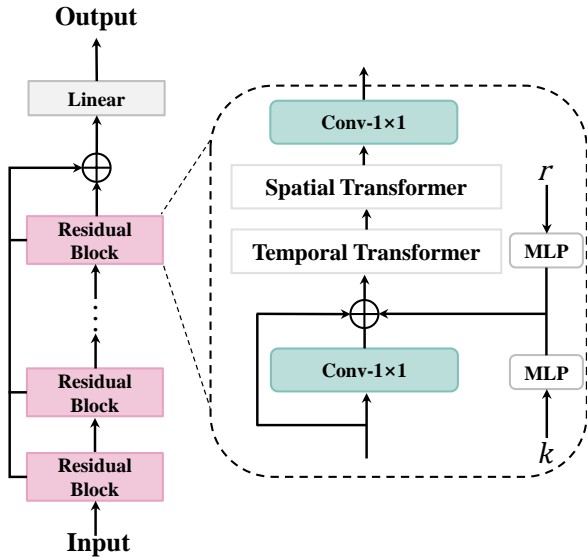

Figure 6: Architecture of Time-Invariant Encoder, consisting of multiple stacked Residual Block.

B.6 HYPERPARAMETER SETTINGS

The detailed hyperparameter settings of our MODEM are presented in Tab. 5. Due to the large number of hyperparameters and resource constraints, we use empirical tuning combined with Bayesian selection to determine the parameter combination. The diffusion step $K$ is set to $50$, while the resolution scale $R$ is $4$, indicating the number of steps for the forward process. The sampling step $L$ is set to 20 for the denoising process. $20\%$ of the frequency components are used for decomposition, represented by $m$. Our model incorporates 2 residual blocks and 4 DMTBs. Additionally, a dilation factor of 2 is applied, and our model is equipped with 5 ModernTCN modules, which are temporal convolutional networks that help handle non-stationary time series. For ensemble inference, the upper percentage at final step $\pi_{T,R}$ is set to 0.02, and the ensemble threshold $\zeta$ is set to 10 for all datasets. These hyperparameter values collectively define our model's structure and its ability to process non-stationarity.

B.7 COMPARISON RESULTS USING AFFILIATON-BASED METRICS

To ensure a more comprehensive comparison and provide a more convincing validation of MODEM's effectiveness, we also evaluate the detection performance using affiliation-based metrics (Wang et al., 2024) to avoid the illusion of progress caused by point adjustment (Kim et al., 2022). As shown in Tab. 6, our MODEM achieves the highest F1-scores across all three datasets with high non-stationarity, with an average improvement of $3.86\%$ (from $80.34\%$ to $84.20\%$). This suggests that the cross-resolution correlation enables a robust reconstruction signal in a coarse-to-fine manner, allowing most anomalous points to be effectively distinguished from normal non-stationary behavior.

Table 5: Detailed hyperparameter settings of MODEM.

| Hyperparameter | Value |
|---|---|
| Diffusion step $K$ | 50 |
| Sampling step $L$ | 20 |
| Resolution scale $R$ | 4 |
| Percentage of frequency decomposition $m$ | 20 |
| Number of residual blocks | 2 |
| Number of DMTBs | 4 |
| Dilated Factor | 2 |
| Number of ModernTCN modules | 5 |
| Upper Percentage $\pi_{T,R}$ | 0.02 |
| Ensemble Threshold $\zeta$ | 10 |

Table 6: Comparison of detection performance across three datasets with high non-stationarity using affiliation-based metrics. The best results are bold, and the second-best are underlined.

| Method | PSM | | | SMD | | | SWaT | | | F1* |
|---|---|---|---|---|---|---|---|---|---|---|
| | P | R | F1 | P | R | F1 | P | R | F1 | |
| Sampling | 84.39 | 51.65 | 64.08 | 74.53 | 31.44 | 44.22 | 60.62 | 84.66 | 70.65 | 59.65 |
| ECOD | 74.60 | 33.84 | 46.56 | 73.98 | 16.15 | 26.51 | 97.61 | 11.51 | 20.59 | 31.22 |
| PCA | 92.20 | 37.71 | 53.53 | 83.88 | 40.19 | 54.34 | 63.58 | 72.18 | 67.61 | 58.49 |
| CBLOF | 59.90 | 98.45 | 74.49 | 86.67 | 33.52 | 48.34 | 63.08 | 70.91 | 66.77 | 63.20 |
| IForest | 100.0 | 3.35 | 6.48 | 100.0 | 9.37 | 17.13 | 61.17 | 70.14 | 65.35 | 29.65 |
| LODA | 92.66 | 40.17 | 56.05 | 59.02 | 66.18 | 62.60 | 61.77 | 70.41 | 65.35 | 61.33 |
| VAE | 62.21 | 87.72 | 72.80 | 82.09 | 43.49 | 58.66 | 60.11 | 73.19 | 66.02 | 65.79 |
| DeepSVDD | 74.05 | 50.64 | 60.15 | 64.98 | 46.77 | 54.13 | 59.11 | 72.35 | 65.69 | 59.99 |
| LSTM-AE | 75.11 | 75.86 | 75.48 | 84.96 | 43.49 | 57.53 | 60.18 | 72.19 | 65.64 | 66.22 |
| MTAD-GAT | 79.90 | 60.14 | 68.63 | 85.90 | 67.69 | 75.71 | 65.09 | 77.51 | 71.23 | 71.86 |
| TFAD | 79.14 | 71.63 | 75.20 | 56.32 | 97.88 | 71.49 | 60.38 | 71.96 | 69.53 | 72.07 |
| AnomalyTransformer | 52.01 | 85.04 | 64.55 | 100.0 | 3.19 | 6.19 | 55.41 | 59.94 | 57.59 | 42.78 |
| D³R | 62.94 | 96.19 | 76.09 | 77.15 | 99.26 | 86.82 | 72.06 | 85.29 | 78.12 | 80.34 |
| MODEM | 73.48 | 87.55 | **79.90** | 89.18 | 95.82 | **92.38** | 74.36 | 87.32 | **80.32** | **84.20** |

## B.8 ADDITIONAL EFFECTIVENESS ANALYSIS

**Multi-Resolution Modeling** We present several case studies in Fig. 7, which quantitatively validate the effectiveness of multi-resolution modeling for anomaly detection. In the figure, the pink and yellow areas represent true anomalies and false anomalies (normal data), respectively. The green and purple lines correspond to the input and reconstructed time series, respectively, with the red bold frame highlighting anomalies detected by the model. These examples demonstrate that models operating at a single resolution struggle to detect true anomalies and frequently trigger false alarms. Benefiting from an expansion to multi-resolution settings, our model learns normal temporal patterns more effectively, providing a more reliable reconstructed baseline that significantly reduces false alarms while accurately detecting true anomalies.

**Frequency-Enhanced Decomposable Network** Previous works (Yang & Hong, 2022; He et al., 2023) have demonstrated that spectral responses can more robustly capture underlying temporal patterns compared to time-domain representations. Despite this foundation, our approach introduces several distinct contributions and innovations for non-stationary time series anomaly detection.

Firstly, we explore different frequency computation strategies and utilize the Short-Term Fourier Transform (STFT) to extract the spectral information. STFT not only provides a detailed view of how the frequency content of the signal changes over time but also captures the transient behavior and dynamics of time series through overlapping windows. Compared to the conventional Fast Fourier Transform (FFT) and the Discrete Fourier Transform (DFT), STFT is more suitable for decomposing non-stationary time series, which is quantitatively verified through experiments on PSM and SMD datasets, as shown in Tab. 7.

Secondly, the proposed Frequency-Enhanced Decomposable Network differs significantly in its network structure from the MLP-based block like Koopa (Liu et al., 2024). Specifically, it incorporates

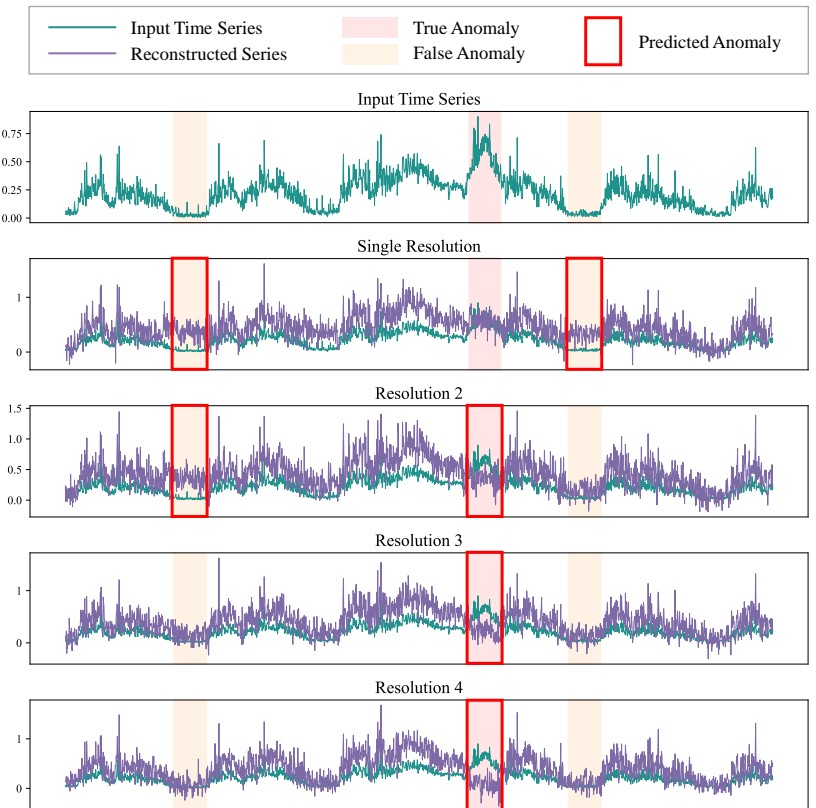

Figure 7: Comparison of detection performance across different resolution settings originates from the SMD dataset, where the pink and yellow areas represent true anomalies and false anomalies (normal data), respectively. The green and purple lines indicate the input time series and the reconstructed time series, respectively, while the red bold frame highlights anomalies detected by the model.

hierarchical transformers in the time-variant and time-invariant encoders to capture both intra-series temporal dependencies and inter-series correlations among variables. Additionally, it designs a novel ModernTCN block enhanced by dilated convolution (named DMTBs) to uncover complex periodic patterns across multiple time scales. These unique designs, compared to Koopa's all-MLP architecture, are more suitable for long-series reconstruction, thus providing reliable reconstruction signals for anomaly detection, as demonstrated by the quantitative validations in Tab. 8.

Table 7: Comparison of different frequency statistics approaches on PSM and SMD datasets.

| Dataset | Approach | P | R | F1 |
|---------|----------|-------|-------|-------|
| PSM | FFT | 96.64 | 97.98 | 97.30 |
| | DFT | 95.86 | 97.12 | 96.48 |
| | STFT | 96.97 | 98.35 | 97.65 |
| SMD | FFT | 95.46 | 96.07 | 95.76 |
| | DFT | 94.73 | 95.44 | 95.08 |
| | STFT | 95.70 | 96.32 | 96.01 |

Table 8: Comparison of different network types on PSM and SMD datasets.

| Dataset | Network | P | R | F1 |
|---------|---------|-------|-------|-------|
| PSM | Koopa (MLP) | 95.68 | 97.53 | 96.59 |
| | MODEM | 96.97 | 98.35 | 97.65 |
| SMD | Koopa (MLP) | 94.42 | 95.43 | 94.92 |
| | MODEM | 95.70 | 96.32 | 96.01 |

In our method, we select frequency components with the top $m$ percent of amplitudes as stationary factors based on spectral statistics, with the remaining frequency components treated as non-stationary factors. As mentioned in Appendix B.6, we set the percentage $m$ to 20, because the frequencies corresponding to the top 20% of amplitudes account for over 90% of all frequency components, which aligns with the reality where stationary factors are dominant. To investigate the impact of

percentage $m$ on the detection performance further, we conduct a sensitivity analysis on the SMD dataset regarding $m$, as illustrated in Tab. 9. Variations within a reasonable range of $m$ do not cause drastic changes in performance, demonstrating that this frequency-based selection approach is robust. If $m$ is set too low (see $m = 2$) or too high (see $m = 50$), the model's performance significantly decreases because the time-variant and time-invariant variables are not effectively separated.

Table 9: Impact of different percentages $m$ on detection performance.

| Percentage $m$ | Frequency Ratio | P | R | F1 |
|---|---|---|---|---|
| 2 | 66.89% | 95.34 | 95.67 | 95.50 |
| 10 | 85.33% | 95.68 | 96.10 | 96.88 |
| 15 | 88.62% | 95.74 | 96.21 | 95.97 |
| 20 | 91.45% | 95.70 | 96.32 | 96.01 |
| 25 | 93.56% | 95.66 | 96.25 | 95.95 |
| 50 | 98.73% | 95.40 | 95.76 | 95.57 |

## B.9 HYPERPARAMETER EXPERIMENTS

**Effectiveness of Resolution Scale**  Regarding the hyperparameter resolution scale $R$, we have validated its impact on detection performance in the SMD dataset in Section 4.3. From this, we selected the recommended value $R = 4$, which we then generalized to the other four datasets to report performance. To more comprehensively evaluate the effectiveness of the multi-resolution setting and the sensitivity of hyperparameter $R$, we fixed the sampling steps at $L = 20$ to sacrifice a small amount of accuracy in favor of inference efficiency, while adjusting $R$ on the other four datasets to observe performance changes. As shown in Tab. 10, the optimal resolution scale $R$ varies slightly across different datasets. We attribute this variation to differences in anomaly rates and the types of anomaly patterns contained in each dataset, which require tuning $R$ to allow the model to effectively benefit from multi-resolution data. In general, for datasets with a low anomaly rate and many short-duration anomalies (lasting only a few time points), we recommend setting a lower resolution scale $R$ (e.g., 4 for the SMD and SMAP datasets) to ensure that anomalies are not overlooked in the low-resolution data. For datasets with a higher anomaly rate and longer-duration anomalies, a higher $R$ should be used to allow the model to fully leverage the temporal correlations across different resolution scales and capture both normal and anomalous event patterns.

Table 10: Results of various resolution scales on PSM, MSL, SWaT, and SMAP datasets.

| $R$ | PSM | | | MSL | | | SWaT | | | SMAP | | |
|---|---|---|---|---|---|---|---|---|---|---|---|---|
| | P | R | F1 | P | R | F1 | P | R | F1 | P | R | F1 |
| 1 | 94.27 | 96.14 | 95.20 | 87.84 | 86.25 | 87.03 | 86.30 | 89.21 | 87.73 | 87.21 | 95.40 | 91.12 |
| 2 | 95.02 | 96.81 | 95.91 | 89.76 | 87.02 | 88.37 | 87.44 | 90.65 | 89.02 | 88.92 | 96.32 | 91.92 |
| 3 | 96.18 | 97.70 | 96.93 | 90.55 | 87.74 | 89.12 | 88.92 | 92.43 | 90.63 | 90.68 | 96.93 | 93.04 |
| 4 | 96.97 | 98.35 | 97.65 | 91.28 | 88.32 | 89.77 | 89.42 | 93.08 | 91.21 | 89.50 | 97.40 | 92.66 |
| 5 | 97.46 | 98.27 | 97.86 | 91.26 | 88.84 | 90.03 | 89.51 | 93.43 | 91.43 | 88.50 | 97.22 | 92.66 |
| 6 | 97.83 | 98.23 | 98.03 | 90.92 | 88.41 | 89.65 | 89.30 | 93.49 | 91.34 | 87.92 | 97.38 | 92.68 |
| 7 | 97.12 | 97.96 | 97.54 | 88.84 | 87.75 | 88.29 | 88.75 | 92.51 | 90.55 | 87.22 | 96.17 | 91.47 |
| 8 | 96.27 | 97.41 | 96.83 | 88.05 | 86.87 | 87.46 | 86.23 | 90.40 | 88.57 | 86.53 | 95.49 | 90.78 |

**Trade-off between Precision and Recall**  It is well-known that there is an inherent trade-off between precision and recall. As described in Appendix B.4, we can balance this trade-off by adjusting the voting threshold $\zeta$ during the ensemble inference process. Specifically, we ensure that the optimal resolution scale $R$ is maintained across all datasets, and then dynamically vary the value of $\zeta$ to observe the impact on anomaly detection performance. As shown in Tab. 11, our approach does not solely focus on optimizing recall to improve detection performance. Regardless of whether $\zeta$ is increased or decreased (except when $\zeta$ is set to a very low value, such as $\zeta = 2$), our model consistently maintains stable performance, with no significant drop in F1-score. In practical applications, by setting a higher value for $\zeta$, our model reduces false positives and thus achieves higher precision. Conversely, a lower $\zeta$ allows the model to detect more anomalies, leading to a

higher recall. This enables a better balance between precision and recall, tailoring the model to meet different real-world requirements.

Table 11: Results of various voting thresholds for ensemble inference on trade-off between precision and recall.

| $\zeta$ | SMD ($R=4$) | | | SMAP ($R=4$) | | | SWaT ($R=5$) | | |
|---|---|---|---|---|---|---|---|---|---|
| | P | R | F1 | P | R | F1 | P | R | F1 |
| 2 | 92.51 | 98.03 | 95.19 | 87.62 | 98.12 | 92.57 | 87.26 | 95.03 | 90.98 |
| 5 | 94.68 | 97.28 | 95.96 | 88.20 | 97.95 | 92.81 | 88.27 | 94.50 | 91.27 |
| 8 | 95.19 | 96.77 | 95.98 | 88.78 | 97.71 | 93.02 | 88.90 | 93.87 | 91.31 |
| 10 | 95.70 | 96.32 | 96.01 | 89.05 | 97.40 | 93.04 | 89.51 | 93.43 | 91.43 |
| 13 | 96.15 | 95.93 | 96.03 | 89.63 | 96.83 | 93.09 | 90.16 | 92.93 | 91.52 |
| 17 | 96.82 | 95.08 | 95.94 | 90.07 | 96.29 | 93.07 | 90.94 | 92.01 | 91.47 |
| 20 | 97.27 | 94.76 | 95.99 | 90.81 | 95.27 | 92.98 | 92.07 | 91.05 | 91.55 |
| 25 | 97.62 | 94.18 | 95.86 | 92.10 | 93.70 | 92.89 | 92.95 | 90.08 | 91.49 |

**Number of Network Layer**  As shown in Fig. 3 and Fig. 6, our time-variant and time-invariant encoders are primarily composed of the proposed Dilated ModernTCN Blocks (DMTB) and residual blocks, respectively. We next explore how varying the number of DMTBs and residual blocks affects anomaly detection performance, providing practical insights for fine-tuning network architecture.

We begin by investigating the relationship between the number of DMTBs and detection performance, as illustrated in Tab. 12. Keeping other hyperparameters constant, we start by setting the number of DMTBs to 0, which means that only linear layers and hierarchical Transformers are used to model the non-stationarity. In this configuration, the model performs the worst on both datasets. As the number of DMTBs increases from 0 to 4, the model's detection performance steadily improves, reaching its highest point. Specifically, the F1-score on the SMD and SWaT datasets increased by 1.05% (from 95.01 to 96.01) and 1.29% (from 90.05 to 91.21), respectively. However, when the number of DMTBs is further increased beyond 4, the performance no longer improves and instead slightly declines on both datasets. This suggests that the benefit of decoupling the time-invariant and time-variant components from complex non-stationary temporal patterns, based on spectral information, diminishes with excessive use of DMTBs. Overusing DMTBs may lead to overfitting, where normal non-stationarity is misidentified as anomalous behavior.

Table 12: Results of various number of Dilated ModernTCN Blocks (DBTMs) on SMD and SWaT datasets.

| Number of DMTBs | SMD | | | SWaT | | |
|---|---|---|---|---|---|---|
| | P | R | F1 | P | R | F1 |
| 0 | 94.59 | 95.44 | 95.01 | 88.30 | 91.86 | 90.05 |
| 1 | 94.82 | 95.75 | 95.28 | 88.82 | 92.45 | 90.60 |
| 2 | 95.15 | 95.93 | 95.54 | 89.16 | 92.73 | 90.91 |
| 3 | 95.48 | 96.21 | 95.84 | 89.30 | 92.94 | 91.08 |
| 4 | 95.70 | 96.32 | 96.01 | 89.42 | 93.08 | 91.21 |
| 5 | 95.64 | 96.28 | 95.95 | 89.33 | 93.16 | 91.20 |
| 6 | 95.52 | 96.04 | 95.78 | 89.17 | 92.85 | 90.97 |

The residual blocks in the time-invariant encoder are designed to capture global dynamic information. As illustrated in Tab. 13, since time-invariant patterns tend to be simpler than time-varying components, we found that increasing the number of residual blocks to 2 yielded satisfactory detection performance. Increasing the number beyond 2 did not result in significant performance gains but did incur additional computational cost. Therefore, we decided to set the number of residual modules in the time-invariant encoder to 2.

### B.10 MODEL EFFICIENCY

We provide a comparison of various time series anomaly detection methods, including both diffusion-based and non-diffusion methods, in terms of computational cost and efficiency on the SMD dataset.

Table 13: Results of various number of Residual Blocks on SMD and SWaT datasets.

| Number of Residual Blocks | SMD | | | SWaT | | |
|---|---|---|---|---|---|---|
| | P | R | F1 | P | R | F1 |
| 0 | 94.38 | 95.54 | 94.96 | 88.54 | 92.45 | 90.45 |
| 1 | 95.37 | 96.16 | 95.76 | 88.97 | 93.01 | 90.94 |
| 2 | 95.70 | 96.32 | 96.01 | 89.42 | 93.08 | 91.21 |
| 3 | 95.62 | 96.26 | 95.94 | 89.56 | 92.98 | 91.23 |
| 4 | 95.54 | 96.12 | 95.83 | 89.22 | 92.90 | 91.02 |

The results are summarized in Tab. 14. Specifically, Training Time represents the time taken to train the model for 5 epochs with the same batch size, while Inference Time indicates the duration required to process the entire test dataset. Additionally, we report the total number of trainable parameters of the model and the FLOPs (floating-point operations per second) per input unit. As observed, diffusion-based models naturally require longer training and inference times due to the iterative nature of solving each diffusion step. However, considering that the training and test data on the SMD dataset spans approximately 16 days, the extra overhead introduced by the diffusion process is acceptable given the significant performance gains it provides for real-world applications. Furthermore, training within the noise space enables diffusion-based methods to detect anomalous data more swiftly, as reflected in their lower ADD.

Even with multi-resolution time series modeling, our MODEM method outperforms the state-of-the-art method D3R, saving $9.62\%$ in training time, $22.8\%$ in inference time, and $82.77\%$ in memory usage. These gains are primarily due to the lighter denoising network and the mathematical extension of the multi-resolution diffusion process to DDIM sampling, which significantly reduces the number of sampling iterations. Regarding the computational overhead of our MODEM, it incorporates hierarchical transformers, and the calculation of attention matrices contributes to higher FLOPs. However, this is acceptable, given the support of highly parallelized attention mechanisms and the continuous advancement of hardware resources.

Moreover, the ongoing development of fast sampling techniques in the diffusion domain, such as DPM Solver++ and single-step sampling, ensures that our multi-resolution paradigm can be further optimized and scaled for wider adoption.

Table 14: Comparison of various methods with respect to training time, inference time, parameters, FLOPs, and ADD.

| Method | Training Time (s) | Inference Time (s) | Total Params (MB) | FLOPs (M) | ADD |
|---|---|---|---|---|---|
| MTAD-GAT | 213.92 | 76.54 | 0.98 | 4.46 | $90 \pm 10$ |
| TFAD | 353.26 | 46.82 | 1.56 | 5.20 | $52 \pm 7$ |
| Anomaly Transformer | 487.85 | 97.24 | 23.28 | 106.38 | $31 \pm 2$ |
| TranAD | 256.77 | 44.60 | 1.97 | 8.56 | $24 \pm 0$ |
| DiffAD | 620.19 | 153.76 | 15.38 | 93.30 | $48 \pm 3$ |
| ImDiffusion | 442.57 | 102.35 | 8.23 | 55.59 | $24 \pm 1$ |
| D3R | 527.48 | 164.28 | 68.27 | 303.27 | $65 \pm 16$ |
| MODEM (ours) | 476.73 | 126.79 | 11.76 | 76.65 | $27 \pm 2$ |

## B.11 GENERALIZATION PERFORMANCE

To assess the generalization ability of MODEM, we conduct forecasting and imputation experiments across multiple non-stationary datasets. We apply MODEM in an unconditional manner to non-stationary time series forecasting and imputation following (Kollovieh et al., 2024), and report the average *continuous ranked probability scores* (CRPS) (Gneiting & Raftery, 2007) across three independent runs in Tab. 15 and Tab. 16, as well as the detailed discussion on experimental setup can be found in Appendix B). The results demonstrate that MODEM performs competitively against SOTA methods, despite not being specifically designed for forecasting tasks. We also extend the proposed multi-resolution diffusion paradigm to CSDI (Tashiro et al., 2021) and TSDiff (Kollovieh et al., 2024) (denoted as CSDI-MR and TSDiff-MR). The improved forecasting and imputation performance validate that correlations across different time scales facilitate the model to better capture diverse temporal patterns.

Table 15: Comparison of forecasting performance on three datasets.

| Method | Traffic | Exchange | KDDCup |
|--------|---------|----------|--------|
| CSDI | $0.159 \pm 0.002$ | $0.033 \pm 0.014$ | $0.318 \pm 0.002$ |
| CSDI-MR | $0.148 \pm 0.004$ | $0.029 \pm 0.008$ | $0.306 \pm 0.004$ |
| TSDiff | $0.098 \pm 0.002$ | $0.011 \pm 0.001$ | $0.311 \pm 0.026$ |
| TSDiff-MR | $0.090 \pm 0.003$ | $0.006 \pm 0.000$ | $0.295 \pm 0.018$ |
| MODEM | $0.096 \pm 0.002$ | $0.013 \pm 0.002$ | $0.305 \pm 0.022$ |

Table 16: Comparison of imputation performance on two datasets.

| Method | Traffic | Exchange |
|--------|---------|----------|
| TSDiff | $0.120 \pm 0.008$ | $0.015 \pm 0.001$ |
| TSDiff-MR | $0.115 \pm 0.015$ | $0.011 \pm 0.001$ |
| MODEM | $0.117 \pm 0.007$ | $0.017 \pm 0.001$ |

## C  LIMITATIONS AND FUTURE WORK

We extend the diffusion model into a multi-resolution paradigm, leading directly to an $R$-fold increase in the number of sampling steps, which results in longer training convergence times and prolonged inference durations. While we introduce accelerated sampling strategies for each resolution $R$, these are predicated on the acceptance of loss in precision. In the future, it may prove beneficial to investigate sampling along a deterministic trajectory within mixed resolutions to improve both accuracy and efficiency. Furthermore, the hyperparameters $R$ and $m$ in our proposed frequency-enhanced decomposition network require customization for different datasets, thereby adding to the complexity of parameter tuning. Additionally, in the future, it may be advantageous to consider employing dynamic smoothing strategies instead of fixed pooling sizes to better reveal the periodic and varying characteristics of non-stationary time series.

## D  POTENTIAL NEGATIVE SOCIETAL IMPACTS

The diffusion models, like other generative technologies, have inherent risks. Our model, as a case in point, could potentially have negative societal effects. For instance, it might memorize private data and be exploited to fabricate misleading or false information.

