# OpenReview forum: "Multi-Resolution Decomposable Diffusion Model for Non-Stationary Time Series Anomaly Detection"
_ICLR.cc/2025/Conference — ICLR 2025 Poster_

### Official Review · Reviewer_VctK · 2024-10-16

**Soundness:** 2
**Presentation:** 3
**Contribution:** 2
**Rating:** 6
**Confidence:** 3

**Summary:**

This paper introduces MODEM, a conditional diffusion model for non-stationary time series anomaly detection that incorporates (1) frequency-enhanced decomposable network, (2) multi-resolution guidance and (3) contextualized (cross-scale contextual) diffusion. The underlying network consists of two parallel encoders using residual Transformer and dilated ModernTCN blocks respectively, while a frequency decomposition module is built to disentangle time-invariant and time-variant components. Empirical studies on a variety of datasets show that this coarse-to-fine diffusion framework achieves good reconstruction performance and captures the key information in the non-stationary data, while ablations show that the specific design of the MODEM outperforms reasonable alternatives.

**Strengths:**

- The paper is well-structured and the main ideas are clearly presented. The figures and supplementary details are relevant and informative.
- To my best knowledge, this is the first work to introduce a multi-granularity mechanism into time series anomaly detection. The same applies to the contextualized forward and reverse conditional diffusion processes.
- The approach is studied through a broad set of experiments that show it consistently outperforms recent, relevant competitor methods in terms of some standard quality metrics for both real-world and non-stationary time series anomaly detection. The additional experiments are conducted, e.g. multivariate time series forecasting and imputation, despite being relatively weak.
- This paper also provides a series of ablation experiments to test the effect of each important choice.

**Weaknesses:**

1. The novelty of the proposal is somehow modest overall. The main body of the proposed method is like a stack of existing works (framework $\rightarrow$ CONTEXTDIFF [1], Top-K iFFT $\rightarrow$ ETSFormer [2], network $\rightarrow$ ModernTCN [3] and D$^3$R [4], multi-resolution $\rightarrow$ MG-TSD [5]), although applying algorithms from different tasks (text-to-image, prediction) to a new task (anomaly detection) can also be seen as a valid contribution.
2. The content of Section 3.3 with theoretical  derivation is very similar to CONTEXTDIFF [1], where the difference is just *coarse-resolution sequence $\leftarrow$ text (condition)* and *high-resolution sequence $\leftarrow$ image (target)*. However, the authors did not clearly tell this point in their paper where the proper citation is not given. In particular, the entire paper cites [1] in `Section 3.3 Multi-Resolution Forward Process` for only once (in an inconspicuous place), and [1] is completely omitted in the theoretical results in Appendix A. I understand that these sections make the paper more solid, but I suggest the authors emphasize contributions of the prior work when necessary.
3. Although important works in the field of anomaly detection are analyzed in Section 2, there are some other diffusion-based approaches that should be included, such as related conditional diffusion model [1], diffusion model for time series [5,6] and so on.

**References:**
> [1] Yang, Ling, et al. "Cross-modal contextualized diffusion models for text-guided visual generation and editing."
[2] Woo, Gerald, et al. "Etsformer: Exponential smoothing transformers for time-series forecasting."
[3] Luo, Donghao, and Xue Wang. "Moderntcn: A modern pure convolution structure for general time series analysis."
[4] Wang, Chengsen, et al. "Drift doesn't matter: dynamic decomposition with diffusion reconstruction for unstable multivariate time series anomaly detection."
[5] Fan, Xinyao, et al. "MG-TSD: Multi-Granularity Time Series Diffusion Models with Guided Learning Process."
[6] Shen, Lifeng, et al. "Multi-Resolution Diffusion Models for Time Series Forecasting."

**Questions:**

1. See the "Weaknesses" section.
2.  From `Section 3.2 Multi-Resolution Data Sampler`: "This strategy enables our model to learn the evolutionary characteristics of non-stationary time series across multiple resolutions, ... "  $\rightarrow$ This statement describes multi-resolution as the contributor of non-stationary time series learning, but how does this facilitation occur ?
3. The computational cost is a concern of diffusion models, especially under real-time anomaly detection. Therefore, comparison of the efficiency of different methods (diffusion and non-diffusion) is of great interest. Can the authors provide more accurate estimates w.r.t. benchmarks within a reasonable time and parameters ?
4. As far as I know, TSDiff was developed for univariate time series generation. Can the authors give more details of your changes in its original paper ?

---

> ### Author Response · Authors · 2024-11-27
> **Rebuttal by Authors 1**
>
> Dear Reviewer VctK:
>
> We sincerely appreciate your time and effort in reviewing our manuscript and for providing insightful comments, which have been incorporated into the updated version. Below we would like to address each of your concerns in detail.
>
> **W1: Novelty of our work.**
>
> We commend your extensive knowledge and appreciate your interest in previous works, which have made impressive achievements across various tasks. Indeed, we have been inspired by some of the methods you mentioned, but we have optimized and innovated these techniques by thoroughly analyzing the characteristics and challenges of non-stationary time series anomaly detection tasks. The objectives and implementation details of our work differ in several key aspects from previous studies. We hope this distinction clarifies the novelty and contributions of our approach. Below, we will provide a detailed comparison with these methods to address your concerns and further clarify our work:
>
> First, in terms of framework, unlike ShiftDDPM and ContextDiff, which utilize classifiers and cross-modal context to facilitate the generation of images or audio, our MODEM emphasizes the explicit optimization of the diffusion and denoising trajectories through the temporal correlations across multiple resolution scales. This allows our model benefit from both general trend information provided by low-resolution data and the detailed event information from high-resolution data. To the best of our knowledge, this is the first attempt to explore multi-resolution diffusion for anomaly detection in non-stationary time series.
>
> Second, in terms of the multi-resolution diffusion setup, MG-TSD models the denoising process as the prediction refinement from the lowest to the highest resolution, designing a multi-granularity guidance loss to allow the model toward higher-resolution data at each diffusion step. However, this technique is not suitable for time series anomaly detection for several reasons. Firstly, relying solely on single-step denoising for cross-scale reconstruction introduces significant errors, especially in non-stationary time series, leading to incorrect reconstruction/anomaly signals. Secondly, very low-resolution data tends to filter out short-term anomalies, which hampers the recall of all types of anomalies. In contrast, our approach innovates by enabling the sequential reconstruction of both lower and higher-resolution time series across all diffusion steps, effectively integrating reconstruction signals across multiple resolutions to better distinguish between normal and anomalous points. Additionally, we explicitly optimize the diffusion and denoising trajectories by leveraging cross-resolution temporal correlations, ensuring that low-resolution temporal patterns propagate into the high-resolution reconstruction.
>
> Third, regarding the series decomposition, ETSFormer extracts the Top-K amplitude components from a single input to separate the seasonal and trend components. However, this approach is not well-suited for separating the time-invariant and time-variant components in non-stationary time series. In such data, the time-varying characteristics complicates the decoupling of these two components. To address this, we instead compute the average spectral statistics across a large set of non-stationary training data, providing a more robust basis for decoupling the time-invariant and time-variant components. The dominant high-amplitude frequency components represent the stationary factors of the time-invariant component, while the remaining low-amplitude frequency components capture the short-term fluctuations or non-stationary factors induced by the time-varying characteristics. Moreover, we utilize the Short-Term Fourier Transform (STFT) to extract the spectral information. STFT not only provides a detailed view of how the frequency content of the signal changes over time but also captures the transient behavior and dynamics of time series through overlapping windows. Compared to traditional Fast Fourier Transform (FFT) and Discrete Fourier Transform (DFT), which do not account for temporal characteristics, STFT is more suitable for non-stationary time series. We provide quantitative evaluation results on PSM and SMD datasets for this in the following table and Table 7 of the updated manuscript (see lines 1179-1186 and lines 1227-1237).
>
> | Dataset | Approach |   P     | R     | F1    |
> |:-:|:-:|:-:|:-:|:-:|
> |         | FFT      | 96.64 | 97.98 | 97.30 |
> |  PSM  | DFT      | 95.86 | 97.12 | 96.48 |
> |         | STFT     | 96.97 | 98.35 | 97.65 |
> |     | FFT      | 95.46 | 96.07 | 95.76 |
> | SMD  | DFT      | 94.73 | 95.44 | 95.08 |
> |         | STFT     | 95.70 | 96.32 | 96.01 |

---

> ### Author Response · Authors · 2024-11-27
> **Rebuttal by Authors 2**
>
> Finally, in terms of network design, we did not draw from D3R due to its heavy network parameters (see lines 1361-1395 and Table 14 for comparison of computational efficiency) and the fact that the diffusion network is only one component of the system. In contrast, our denoising network is primarily divided into two branches: time-variant and time-invariant encoders. It incorporates hierarchical transformers in the time-variant and time-invariant encoders to capture both intra-series temporal dependencies and inter-series correlations among variables. Additionally, it designs a novel ModernTCN block enhanced by dilated convolution (named DMTBs) to uncover complex periodic patterns across multiple time scales. These unique designs, compared to D3R and other method’s all-MLP architecture, are more suitable for long-series reconstruction, thus providing robust reconstruction signals for anomaly detection, as demonstrated by the quantitative validations in the Table 1, Table 6 and Table 8.
>
> We sincerely hope that this will address your concerns and that the novelty and contributions of our work will be reconsidered.
>
> **W2: Emphasis on contributions of prior work.**
>
> Thank you for your valuable feedback, which helps clarify the contributions of our work. In response, we have clearly outlined the inspiration drawn from ShiftDDPM and ContextDiff for our multi-resolution diffusion theory, as well as the key distinctions between our approach and theirs, in Section 3.3 (lines 236-240, highlighted in green) of the updated manuscript, as detailed below:
>
> "Inspired by, but distinct from, ShiftDDPM Zhang et al. (2023) and ContextDiff Yang et al. (2024), which utilize classifiers and cross-modal contextual information respectively, we are the first to leverage the temporal correlations across multiple resolution scales to modify the diffusion trajectory, ensuring that lower-resolution data effectively facilitate the reconstruction of higher-resolution data."
>
> Moreover, in the mathematical derivations of Appendix A, we have emphasized the inspiration and contributions (highlighted in green) from prior works to our multi-resolution diffusion model.
>
> **W3: Discussions on some other diffusion-based approaches.**
>
> Thank you for your reminder. We have discussed and compared some other diffusion-based approaches in Section 2 and Section 3.3 of the updated manuscript to clarify the contributions of our work. Specifically, please refer to the discussion on the importance of multi-resolution techniques in the time series domain in Section 2 (see lines 153-158, highlighted in green):
>
> "Notably, the importance of multi-resolution techniques Li et al. (2023a); Zhang et al. (2024) has been demonstrated in time series forecasting. For instance, MG-TSD Fan et al. (2024) designs a multi-granularity guidance loss to guide the learning process of diffusion models, while MR-Diff Shen et al. (2024) employs progressive denoising to generate coarser and finer trend signals, improving the accuracy of time series prediction. These works inspire us to be the first to analyze normal and anomalous change patterns in non-stationary data across multiple resolution scales. "
>
> Additionally, the comparison with previous works, ShiftDDPM and ContextDiff, can be found in Section 3.3 of the updated manuscript (see lines 236-240, highlighted in green).
>
> **Q1: Response to the ‘Weaknesses’ section.**
>
> We have provided detailed responses to each of the weaknesses you raised above and have updated the manuscript accordingly. We hope these responses and revisions address your concerns satisfactorily.
>
> **Q2: Facilitation of the multi-resolution setting.**
>
> The multi-resolution data sampler samples time series at multiple resolution scales $r$ from high-resolution data using average pooling operation. This strategy provides the necessary data support for our model to perform multi-resolution learning, ensuring that our model captures the evolutionary characteristics of non-stationary time series across multiple resolutions during the diffusion and denoising processes. To avoid any confusion, we have revised the statement of the multi-resolution data sampler's function as follows:
>
> "This strategy provides the necessary data support for our model’s subsequent multi-resolution learning, enabling it to capture evolutionary characteristics of non-stationary time series across multiple resolutions and enhancing its understanding of normal and anomalous change patterns."
>
> These clarifications have been incorporated into the updated version of the manuscript in Section 3.2 (see lines 201-204, highlighted in green).

---

> ### Author Response · Authors · 2024-11-27
> **Rebuttal by Authors 3**
>
> **Q3: Comparison of the efficiency of different methods.**
>
> Thank you for your suggestion. We have provided a comparison of various time series anomaly detection methods, including both diffusion-based and non-diffusion methods, in terms of computational cost and efficiency on the SMD dataset. The results are summarized in the table below. Specifically, Training Time represents the time taken to train the model for $5$ epochs with the same batch size, while Inference Time indicates the duration required to process the entire test dataset. Additionally, we report the total number of trainable parameters of the model and the FLOPs per input unit.
>
> |Method|Training Time (s)|Inference Time (s)|Total Params (MB)|FLOPs (M)|ADD|
> |--|:-:|:-:|:-:|:-:|-|
> |MTAD-GAT|213.92|76.54|0.98|4.46|90 $\pm$ 10|
> |TFAD|353.26|46.82|1.56|5.20|52 $\pm$ 7|
> |AnomalyTransformer|487.85|97.24|23.28|106.38|31 $\pm$ 2|
> |TranAD|256.77|44.60|1.97|8.56|24 $\pm$ 0|
> |DiffAD|620.19|153.76|15.38|93.30|48 $\pm$ 3|
> |ImDiffusion|442.57|102.35|8.23|55.59|24 $\pm$ 1|
> |D3R|527.48|164.28|68.27|303.27|65 $\pm$ 16|
> |MODEM(ours)|476.73|126.79|11.76|76.65|27 $\pm$ 2|
>
> As observed, diffusion-based models naturally require longer training and inference times due to the iterative nature of solving each diffusion step. However, considering that the training and test data on the SMD dataset spans approximately $16$ days, the extra overhead introduced by the diffusion process is acceptable given the significant performance gains it provides for real-world applications. Furthermore, training within the noise space enables diffusion-based methods to detect anomalous data more swiftly, as reflected in their lower ADD.
>
> Even with multi-resolution time series modeling, our MODEM method outperforms the state-of-the-art method D3R, saving $9.62\%$ in training time, $22.8\%$ in inference time, and $82.77\%$ in memory usage. These gains are primarily due to the lighter denoising network and the mathematical extension of the multi-resolution diffusion process to DDIM sampling, which significantly reduces the number of sampling iterations.
>
> Regarding the computational overhead of our MODEM, it incorporates hierarchical transformers, and the calculation of attention matrices contributes to higher FLOPs. However, this is acceptable, given the support of highly parallelized attention mechanisms and the continuous advancement of hardware resources. Moreover, the ongoing development of fast sampling techniques in the diffusion domain, such as DPM Solver++ and single-step sampling, ensures that our multi-resolution paradigm can be further optimized and scaled for wider adoption.
>
> The above comparison results and discussions on computational efficiency have been included in Section 4.3 (see lines 395-401, highlighted in blue) and Appendix B.10 (see lines 1361-1395, highlighted in blue) of the updated manuscript.
>
> **Q4: Explanation of the changes to TSDiff.**
>
> TSDiff is developed for univariate time series generation, and the selected datasets—Traffic, Exchange, and KDDCup—are widely used, preprocessed univariate datasets derived from traffic, financial exchange rates, and climate domains, respectively. These datasets, after processing, can be obtained from Appendix B.1 of the TSDiff paper. When extending the proposed multi-resolution approach to TSDiff, we preserved its original network structure and hyperparameter settings as presented in the original paper and code. The only additions were the resolution scale $R$ and an extra cross-attention layer to capture correlations across different resolutions. In practical implementation, we modified the original diffusion and sampling processes so that TSDiff first generates low-resolution data and then refines it to high-resolution data.
>
> Moreover, similar to TSDiff, our proposed MODEM is also an unconditional diffusion model. When applying MODEM to univariate datasets, the Spatial Transformer components in both the time-invariant and time-variant encoders were omitted, and the model's input dimensions were adjusted based on the specific context length and prediction length of each dataset. The resolution scale, context length, and prediction length for the Traffic, Exchange, and KDDCup datasets are set to $(3, 336, 24)$, $(3, 360, 30)$, and $(4, 312, 48)$, respectively. These modifications ensure seamless deployment of MODEM on univariate data.
>
> These clarifications have been incorporated into the updated version of the manuscript in Appendix B.2 (see lines 1006-1021, highlighted in green). Additionally, we will include the above generalization performance evaluation in the released code.
>
> We hope the above responses have adequately addressed your concerns, and we would greatly appreciate it if you could adjust your rating accordingly. If you still have any remaining questions or concerns about our work, please do not hesitate to let us know, and we will respond promptly to resolve them.
>
> Best regards,
>
> Authors

---

> > ### Comment · Reviewer_VctK · 2024-11-28
> >
> > Thank the authors for the extensive response. I think most of points and my questions were addressed by the updated submission. I have thus updated my assessment to reflect the changes made to the paper.

---

> > > ### Author Response · Authors · 2024-11-28
> > >
> > > We sincerely appreciate your constructive pre-rebuttal review and your decision to raise the rating.
> > >
> > > Should you have any remaining concerns or unresolved questions regarding our work, please do not hesitate to inform us. Your valuable feedback will greatly assist us in further improving our manuscript.
> > >
> > > Thank you once again for your time and insightful comments.

---

### Official Review · Reviewer_hGSS · 2024-10-23

**Soundness:** 3
**Presentation:** 3
**Contribution:** 3
**Rating:** 6
**Confidence:** 3

**Summary:**

The paper introduces a novel Multi-Resolution Decomposable Diffusion Model (MODEM), designed for unsupervised non-stationary time-series anomaly detection. It addresses the challenge of capturing complex temporal patterns in non-stationary data by leveraging multi-resolution modeling. MODEM employs a coarse-to-fine diffusion process to handle cross-resolution correlations, improving anomaly detection and time-series reconstruction. A frequency-enhanced decomposable network in MODEM extracts both global and time-variant dynamics. Extensive experiments across five real-world datasets demonstrate its superior performance.

**Strengths:**

The paper is well-organized with clear writing and visualization. The experiments are comprehensive.

**Weaknesses:**

1.	Considering the FFT process and the multi-resolution, it may cast a concern on the training time and the computational complexity, which are not sufficiently discussed in the paper, since ADD seems to be a metric for responsiveness of the method, not its computational efficiency.

2.	In addition to reconstructed-based and forecasting-based methods, there are also some density-based methods can be discussed [1,2].

Reference:

[1] Graph-Augmented Normalizing Flows for Anomaly Detection of Multiple Time Series

[2] Detecting Multivariate Time Series Anomalies with Zero Known Label

**Questions:**

1.	Regarding the ablation study, for SMD, increasing R from 1 to 4 improves the performance in Figure 4. Does this generalize to other datasets? For datasets with many short-lasting anomalies (spanning only a few time points), would these anomalies only be captured in high-resolution data? If so, the utility of low-resolution data in such cases remains questionable.

2.	Is the time series in Figure 1 are real-world data? If so, which domain does the data belong to, and do similar time series challenges appear across other domains?

---

> ### Author Response · Authors · 2024-11-27
> **Rebuttal by Authors 1**
>
> Dear Reviewer hGSS:
>
> We sincerely appreciate your time and effort in reviewing our manuscript and for providing insightful comments, which have been incorporated into the updated version. Below we would like to address each of your concerns in detail.
>
> **W1: Discussions on the training time and the computational efficiency.**
>
> Thank you for your suggestion. We have provided a comparison of various time series anomaly detection methods, including both diffusion-based and non-diffusion methods, in terms of computational cost and efficiency on the SMD dataset. The results are summarized in the table below. Specifically, Training Time represents the time taken to train the model for $5$ epochs with the same batch size, while Inference Time indicates the duration required to process the entire test dataset. Additionally, we report the total number of trainable parameters of the model and the FLOPs (floating-point operations per second) per input unit.
>
> | Method  | Training Time (s) | Inference Time (s) | Total Params (MB) | FLOPs (M) | ADD  |
> |--|:-:|:-:|:-:|:-:|:-:|
> | MTAD-GAT | 213.92  | 76.54 | 0.98  | 4.46   | 90 $\pm$ 10  |
> | TFAD  | 353.26  | 46.82  | 1.56 | 5.20  | 52 $\pm$ 7   |
> | Anomaly Transformer  | 487.85 | 97.24   | 23.28  | 106.38    | 31 $\pm$ 2   |
> | TranAD  | 256.77 | 44.60  | 1.97 | 8.56     | 24 $\pm$ 0   |
> | DiffAD  | 620.19 | 153.76   | 15.38    | 93.30    | 48 $\pm$ 3   |
> | ImDiffusion | 442.57  | 102.35  | 8.23  | 55.59    | 24 $\pm$ 1   |
> | D3R | 527.48  | 164.28  | 68.27   | 303.27    | 65 $\pm$ 16  |
> | MODEM (ours)  | 476.73  | 126.79  | 11.76   | 76.65     | 27 $\pm$ 2   |
>
> As observed, diffusion-based models naturally require longer training and inference times due to the iterative nature of solving each diffusion step. However, considering that the training and test data on the SMD dataset spans approximately $16$ days, the extra overhead introduced by the diffusion process is acceptable given the significant performance gains it provides for real-world applications. Furthermore, training within the noise space enables diffusion-based methods to detect anomalous data more swiftly, as reflected in their lower ADD.
>
> Even with multi-resolution time series modeling, our MODEM method outperforms the state-of-the-art method D3R, saving $9.62\%$ in training time, $22.8\%$ in inference time, and $82.77\%$ in memory usage. These gains are primarily due to the lighter denoising network and the mathematical extension of the multi-resolution diffusion process to DDIM sampling, which significantly reduces the number of sampling iterations.
>
> Regarding the computational overhead of our MODEM, it incorporates hierarchical transformers, and the calculation of attention matrices contributes to higher FLOPs. However, this is acceptable, given the support of highly parallelized attention mechanisms and the continuous advancement of hardware resources. Moreover, the ongoing development of fast sampling techniques in the diffusion domain, such as DPM Solver++ and single-step sampling, ensures that our multi-resolution paradigm can be further optimized and scaled for wider adoption.
>
> The above comparison results and discussions on computational efficiency have been included in Section 4.3 (see lines 395-401, highlighted in blue) and Appendix B.10 (see lines 1361-1395, highlighted in blue) of the updated manuscript.
>
> **W2: Discussions on some density-based methods.**
>
> Thank you for your reminder. We have discussed excellent density-based anomaly detection methods, such as MTGFlow and GANF, in Section 1 and Section 2 of the updated manuscript to provide a more comprehensive review of previous work. For details, please refer to lines 39-42 in Section 1 (highlighted in purple):
>
> "These methods are generally divided into three categories: density-based, forecasting-based and reconstruction-based. Density-based methods (Dai & Chen; Zhou et al., 2023; 2024) detect anomalies based on the assumption that anomalies often lie on low-density regions of data distribution."
>
> Additionally, we have included a discussion on density-based approaches in Section 2 (lines 129-133, highlighted in purple):
>
> "Existing unsupervised anomaly detection methods for time series are primarily categorized into density-based (Dai & Chen; Zhou et al., 2023), forecasting-based (Yao et al., 2022), and reconstruction-based (Zong et al., 2018; Audibert et al., 2020) approaches. Density-based methods focus on fitting the density of training and test samples to detect anomalies, which are based on the assumption that anomalies often lie on low-density regions of data distribution (Zhou et al., 2024)."

---

> ### Author Response · Authors · 2024-11-27
> **Rebuttal by Authors 2**
>
> **Q1: Concerns on the resolution scale $R$.**
>
> In the ablation study, we validated that adjusting the resolution scale $R=4$ achieves the best detection performance on the SMD dataset. We directly extended this suggested value to the other four datasets, as shown in Tables 1 and 6. Benefiting from the explicit optimization of the diffusion trajectory through cross-resolution correlations, our method achieves state-of-the-art performance, demonstrating the generalizability of this hyperparameter setting. For short-term anomalies that span only a few time points, low-resolution data may indeed fail to detect these anomalies because they are filtered out. However, our approach does not solely rely on low-resolution data. Instead, by exploring temporal correlations across different resolution scales, our model benefits from both the general trend information provided by low-resolution data and the detailed event information provided by high-resolution data, allowing for better differentiation between normal and anomalous patterns. Additionally, our ensemble inference strategy integrates the reconstruction signals from both high-resolution and low-resolution data, enabling the model to gather useful information from other resolutions when anomalies are difficult to detect in one resolution.
>
> In summary, within our framework, high-resolution and low-resolution time series complement each other, jointly providing the model with robust reconstruction signals for anomaly detection.
>
> To more comprehensively evaluate the effectiveness of our multi-resolution settings, we adjusted $R$ on the other four datasets to observe performance changes. As shown in the table below, the optimal resolution scale $R$ varies slightly across different datasets. We attribute this variation to differences in anomaly rates and the types of anomaly patterns contained in each dataset, which require tuning $R$ to allow the model to effectively benefit from multi-resolution data. In general, for datasets with a low anomaly rate and many short-duration anomalies (lasting only a few time points), we recommend setting a lower resolution scale $R$ (e.g., 4 for the SMD and SMAP datasets) to ensure that anomalies are not overlooked in the low-resolution data. For datasets with a higher anomaly rate and longer-duration anomalies, a higher $R$ should be used to allow the model to fully leverage the temporal correlations across different resolution scales and capture both normal and anomalous event patterns.
>
> | Resolution Scale | | PSM | | | MSL | | | SWaT | | | SMAP | |
> |:-:|:-:|:-:|:-:|:-:|:-:|:-:|:-:|:-:|:-:|:-:|:-:|:-:|
> | $R$  | P | R | F1| P | R | F1| P | R | F1| P | R | F1| P | R | F1|
> |1 | 94.27 | 96.14 | 95.20 | 87.84 | 86.25 | 87.03 | 86.30 | 89.21 | 87.73 | 87.21 | 95.40 | 91.12 |
> |2 | 95.02 | 96.81 | 95.91 | 89.76 | 87.02 | 88.37 | 87.44 | 90.65 | 89.02 | 87.92 | 96.32 | 91.92 |
> |3 | 96.18 | 97.70 | 96.93 | 90.55 | 87.74 | 89.12 | 88.92 | 92.43 | 90.64 | 88.37 | 96.93 | 92.45 |
> |4 | 96.97 | 98.35 | 97.65 | 91.28 | 88.32 | 89.77 | 89.42 | 93.08 | 91.21 | 89.05 | 97.40 | **93.04** |
> |5 | 97.46 | 98.27 | 97.86 | 91.26 | 88.84 | **90.03** | 89.51 | 93.43 | **91.43** | 88.50 | 97.22 | 92.66|
> |6 | 97.83 | 98.23 | **98.03** | 90.92 | 88.41 | 89.65 | 89.30 | 93.49 | 91.34 | 88.41 | 97.38 | 92.68 |
> |7 | 97.12 | 97.96 | 97.54 | 88.84 | 87.75 | 88.29 | 88.75 | 92.51 | 90.59 | 87.22 | 96.17 | 91.47 |
> |8 | 96.27 | 97.41 | 96.83 | 88.05 | 86.87 | 87.46 | 86.23 | 90.40 | 88.57 | 86.53 | 95.49 | 90.78 |
>
> The above experimental results and discussions on resolution scale $R$ have been included in Section 4.5 (see lines 482-506) and Appendix B.9 (see lines 1262-1288, highlighted in red) of the updated manuscript.
>
>
> **Q2: Confusion about the non-stationary series in Figure 1.**
>
> We guess that you are referring to Figure 1(a). The time series shown in Figure 1(a) is sourced from the real-world SMD dataset (machine-3-4_train.pkl), which contains sensor data from $28$ server machines of a large Internet company. The challenges posed by non-stationary data are also prevalent in other domains. For instance, in the SWaT dataset, environmental changes can lead to performance fluctuations in devices like water pumps and filters, even when they are operating normally. Furthermore, time-varying shifts in water treatment process parameters add to the non-stationary nature of the collected time series. Similarly, in the PSM dataset, dynamic changes in server load, the superposition of multiple user and service patterns, and hardware aging all contribute to the emergence of non-stationary time series.
>
> We hope the above responses have adequately addressed your concerns, and we would greatly appreciate it if you could adjust your rating accordingly. If you still have any remaining questions or concerns about our work, please do not hesitate to let us know, and we will respond promptly to resolve them.
>
> Best regards,
>
> Authors

---

> ### Author Response · Authors · 2024-11-30
> **Request for Reviewer's Attention and Feedback**
>
> Dear Reviewer hGSS:
>
> We would like to express our sincere gratitude for the time and effort you dedicated to reviewing our manuscript, as well as for your insightful and constructive comments. We have carefully considered each of your comments and have provided detailed responses to address the concerns you raised.
>
> As the reviewer-author discussion phase is scheduled to conclude in less than three days, we kindly request that you confirm whether our responses have sufficiently addressed your concerns, and if there are any remaining issues that still require attention. We greatly appreciate your feedback and would be grateful for any further suggestions you may have.
>
> We look forward to your feedback and thank you once again for your continued support. We hope you have a wonderful and restful Thanksgiving holiday!
>
> Best regards,
>
> Authors

---

> > ### Comment · Reviewer_hGSS · 2024-12-01
> >
> > Thank you for your response. My concerns were addressed and I raised my score to 6.

---

> > > ### Author Response · Authors · 2024-12-01
> > >
> > > We sincerely appreciate your constructive pre-rebuttal review and your decision to raise the rating.
> > >
> > > Should you have any remaining concerns or unresolved questions regarding our work, please do not hesitate to inform us. Your valuable feedback will greatly assist us in further improving our manuscript.
> > >
> > > Thank you once again for your time and insightful comments.

---

### Official Review · Reviewer_meLm · 2024-11-02

**Soundness:** 3
**Presentation:** 3
**Contribution:** 3
**Rating:** 6
**Confidence:** 3

**Summary:**

In this paper, authors design a modal named MODEM（Multi-Resolution Decomposable Diffusion Model）for anomaly detection in non-stationary time series data. The proposed model integrates a coarse-to-fine diffusion paradigm with a frequency-enhanced decomposable network to enhance the modal’s ability of handing the non-stationary time series data. Through this approach, the proposed model allows for utilizing the multiple resolutions information of non-stationary time series data, which varies for each datasets.

**Strengths:**

1)The proposed method MODEM is novel , it can utilize the multiple resolutions information of non-stationary time series data to improve the modal’s ability of anomaly detection.

2)The authors conducted experiments on diverse datasets, and their proposed method demonstrated excellent performance.

3)The authors presented a clear exposition of their research motivation and proposed solution. The analyze of difference between low-resolution time series and high-resolution time series helps understand the effect of the proposed modules.

**Weaknesses:**

1)While the model achieved the best F-score performance due to higher recall values, its precision values were comparatively lower than other models (Table 3). The authors should explain this observation.

2)There is no comparison of computational cost(# of params, Gflops...) compared to existing models, while the diffusion model is extended into a multi-resolution paradigm, leading directly to an R-fold increase in the number of sampling steps.

**Questions:**

see Weaknesses

---

> ### Author Response · Authors · 2024-11-27
> **Rebuttal by Authors 1**
>
> Dear Reviewer meLm:
>
> We sincerely appreciate your time and effort in reviewing our manuscript and for providing insightful comments, which have been incorporated into the updated version. Below we would like to address each of your concerns in detail.
>
> **W1: Clarification on the precision-recall tradeoffs.**
>
> It is well-known that there is an inherent trade-off between precision and recall. As described in Appendix B.4, we can balance this trade-off by adjusting the voting threshold $\zeta$ during the ensemble inference process. The performance improvement of our method, particularly in recall, is mainly due to setting a relatively small voting threshold $\zeta=10$, which allows more anomalies to be detected with a more lenient criterion. To verify that our model does not simply optimize recall to enhance detection performance, we varied the voting threshold $\zeta$ across three datasets and observed its impact on anomaly detection performance. As shown in the table below, as $\zeta$ increases, our model reduces false positives and achieves higher precision. However, this also leads to a decrease in recall. Despite this trade-off, our model continues to demonstrate stable performance, as evidenced by only slight fluctuations in F1-score. This indicates that our model effectively balances precision and recall to meet the practical requirements of real-world anomaly detection tasks.
> | Voting Threshold | | SMD ($R=4$) | | | SMAP ($R=4$) | | | SWaT ($R=5$) | |
> |:-:|:-:|:-:|:-:|:-:|:-:|:-:|:-:|:-:|:-:|
> | $\zeta$  | P | R | F1| P | R | F1| P | R | F1|
> |2 | 92.51 | 98.03 | 95.19 | 87.62 | 98.12 | 92.57 | 87.26 | 95.03 | 90.98 |
> |5 | 94.68 | 97.28 | 95.96 | 88.20 | 97.95 | 92.81 | 88.27 | 94.50 | 91.27 |
> |8 | 95.19 | 96.77 | 95.98 | 88.78 | 97.71 | 93.02 | 88.90 | 93.87 | 91.31 |
> |10| 95.70 | 96.32 | 96.01 | 89.05 | 97.40 | 93.04 | 89.51 | 93.43 | 91.43 |
> |13| 96.15 | 95.93 | 96.03 | 89.63 | 96.83 | 93.09 | 90.16 | 92.93 | 91.52 |
> |17| 96.82 | 95.08 | 95.94 | 90.07 | 96.29 | 93.07 | 90.94 | 92.01 | 91.47 |
> |20| 97.27 | 94.76 | 95.99 | 90.81 | 95.27 | 92.98 | 92.07 | 91.05 | 91.55 |
> |25| 97.62 | 94.18 | 95.86 | 92.10 | 93.70 | 92.89 | 92.95 | 90.08 | 91.49 |
>
> The above experimental results and discussions on precision-recall-tradeoff have been included in Appendix B.9 of the updated manuscript, specifically in lines 1291-1314, highlighted in red.

---

> ### Author Response · Authors · 2024-11-27
> **Rebuttal by Authors 2**
>
> **W2: Comparison of the computational cost.**
>
> Thank you for your suggestion. We have provided a comparison of various time series anomaly detection methods, including both diffusion-based and non-diffusion methods, in terms of computational cost and efficiency on the SMD dataset. The results are summarized in the table below. Specifically, Training Time represents the time taken to train the model for $5$ epochs with the same batch size, while Inference Time indicates the duration required to process the entire test dataset. Additionally, we report the total number of trainable parameters of the model and the FLOPs (floating-point operations per second) per input unit.
>
> | Method  | Training Time (s) | Inference Time (s) | Total Params (MB) | FLOPs (M) | ADD  |
> |--|:-:|:-:|:-:|:-:|:-:|
> | MTAD-GAT | 213.92  | 76.54 | 0.98  | 4.46   | 90 $\pm$ 10  |
> | TFAD  | 353.26  | 46.82  | 1.56 | 5.20  | 52 $\pm$ 7   |
> | Anomaly Transformer  | 487.85 | 97.24   | 23.28  | 106.38    | 31 $\pm$ 2   |
> | TranAD  | 256.77 | 44.60  | 1.97 | 8.56     | 24 $\pm$ 0   |
> | DiffAD  | 620.19 | 153.76   | 15.38    | 93.30    | 48 $\pm$ 3   |
> | ImDiffusion | 442.57  | 102.35  | 8.23  | 55.59    | 24 $\pm$ 1   |
> | D3R | 527.48  | 164.28  | 68.27   | 303.27    | 65 $\pm$ 16  |
> | MODEM (ours)  | 476.73  | 126.79  | 11.76   | 76.65     | 27 $\pm$ 2   |
>
> As observed, diffusion-based models naturally require longer training and inference times due to the iterative nature of solving each diffusion step. However, considering that the training and test data on the SMD dataset spans approximately $16$ days, the extra overhead introduced by the diffusion process is acceptable given the significant performance gains it provides for real-world applications. Furthermore, training within the noise space enables diffusion-based methods to detect anomalous data more swiftly, as reflected in their lower ADD.
>
> Even with multi-resolution time series modeling, our MODEM method outperforms the state-of-the-art method D3R, saving $9.62\%$ in training time, $22.8\%$ in inference time, and $82.77\%$ in memory usage. These gains are primarily due to the lighter denoising network and the mathematical extension of the multi-resolution diffusion process to DDIM sampling, which significantly reduces the number of sampling iterations.
>
> Regarding the computational overhead of our MODEM, it incorporates hierarchical transformers, and the calculation of attention matrices contributes to higher FLOPs. However, this is acceptable, given the support of highly parallelized attention mechanisms and the continuous advancement of hardware resources. Moreover, the ongoing development of fast sampling techniques in the diffusion domain, such as DPM Solver++ and single-step sampling, ensures that our multi-resolution paradigm can be further optimized and scaled for wider adoption.
>
> The above comparison results and discussions on computational efficiency have been included in Section 4.3 (see lines 395-401, highlighted in blue) and Appendix B.10 (see lines 1361-1395, highlighted in blue) of the updated manuscript.
>
>
> We hope the above responses have adequately addressed your concerns, and we would greatly appreciate it if you could adjust your rating accordingly. If you still have any remaining questions or concerns about our work, please do not hesitate to let us know, and we will respond promptly to resolve them.
>
> Best regards,
>
> Authors

---

> ### Author Response · Authors · 2024-11-30
> **Request for Reviewer's Attention and Feedback**
>
> Dear Reviewer meLm:
>
> We would like to express our sincere gratitude for the time and effort you dedicated to reviewing our manuscript, as well as for your insightful and constructive comments. We have carefully considered each of your comments and have provided detailed responses to address the concerns you raised.
>
> As the reviewer-author discussion phase is scheduled to conclude in less than three days, we kindly request that you confirm whether our responses have sufficiently addressed your concerns, and if there are any remaining issues that still require attention. We greatly appreciate your feedback and would be grateful for any further suggestions you may have.
>
> We look forward to your feedback and thank you once again for your continued support. We hope you have a wonderful and restful Thanksgiving holiday!
>
> Best regards,
>
> Authors

---

> ### Author Response · Authors · 2024-12-02
> **Request for Reviewer's Attention and Feedback**
>
> Dear Reviewer meLm:
>
> Once again, we sincerely appreciate the time and effort you dedicated to reviewing our manuscript, as well as your insightful comments.
>
> As the discussion period is drawing to a close, we would like to bring your attention, and kindly request that you confirm whether our responses have adequately addressed your concerns. If so, we would be grateful if you could consider adjusting the rating accordingly.
>
> If you have any further suggestions or questions, please do not hesitate to let us know, and we will promptly provide our responses. We eagerly look forward to your feedback and deeply appreciate your continued support.
>
> Best regards,
>
> Authors

---

> ### Author Response · Authors · 2024-12-03
> **Request for Reviewer's Attention and Feedback**
>
> Dear Reviewer meLm:
>
> **With less than 10 hours remaining in the reviewer-author discussion period**, we would like to bring your attention to our responses and revisions made to our manuscript.
>
> We sincerely appreciate the time and effort you dedicated to reviewing our submission, as well as your insightful comments. We have carefully considered each of your concerns and have provided detailed responses.
>
> **We kindly request that you confirm whether our responses have satisfactorily addressed your concerns.** If they have, we would be deeply grateful if you could consider adjusting the rating accordingly.
>
> If you have any further suggestions or questions, please do not hesitate to let us know, and we will promptly provide our clarifications. We eagerly look forward to your feedback and deeply appreciate your continued support.
>
> Best regards,
>
> Authors

---

> ### Author Response · Authors · 2024-12-03
> **Final Request for Reviewer's Attention and Feedback**
>
> Dear Reviewer meLm:
>
> **With only 3 hours remaining in the reviewer-author discussion period**, we would like to kindly make a final request for you to confirm whether our responses have satisfactorily addressed your concerns.
>
> We are truly grateful for your insightful comments and the effort you have dedicated to reviewing our manuscript. We eagerly look forward to your feedback and deeply appreciate your continued support.
>
> Sincerely,
>
> Authors

---

### Official Review · Reviewer_cQvr · 2024-11-04

**Soundness:** 3
**Presentation:** 4
**Contribution:** 3
**Rating:** 8
**Confidence:** 3

**Summary:**

The authors introduce a multiscale diffusion model for anomaly detection in non-stationary time series that relies on combining representations of the time series at multiple temporal resolutions.

**Strengths:**

- The paper is well written, mathematically sound and clearly formulates subcomponents of the method.

- The experimental evaluation and comparisons are rather extensive.

**Weaknesses:**

- My main concern centres around hyperparameter settings for which the authors claim to follow empirical tuning with Bayesian selection without providing further details. I understand the difficulties stated by the authors in properly tuning the large number of hyperparameters; however without an objective method for setting parameters based on training data alone it is not clear whether the selected parameter settings will generalise well to other datasets. Here, the authors should at least provide some analysis of how sensitive the results are to variations of individual parameter values by keeping the rest of the parameters at their suggested values.
- The resolution parameter R is set to a rather low value of 4 in the experiments and hence covers only a narrow range of resolutions which seems contrary to the overall claim of the authors that the method leverages correlations across various resolution scales.
- The method seems to be mainly improving on the recall value while trailing behind interns of precision, sometimes significantly, compared to other methods. I believe further clarification of this tradeoff in practical scenarios is needed.
- The authors should provide more details on the pooling operation in the MRD sampler.
- No code is provided which makes proper assesment of the experimental results and method difficult.

**Questions:**

- see Weaknesses

---

> ### Author Response · Authors · 2024-11-27
> **Rebuttal by Authors 1**
>
> Dear Reviewer cQvr:
>
> We sincerely appreciate your time and effort in reviewing our manuscript and for providing insightful comments, which have been incorporated into the updated version. Below we would like to address each of your concerns in detail.
>
> **W1: Concerns on the hyperparameter settings.**
>
> Thank you for your valuable suggestion. In response to your feedback, we have added extensive experimental results regarding the main hyperparameters introduced in our work in the updated manuscript. Below, we will present the results of each hyperparameter experiment to analyze their sensitivity and generalizability.
>
> First, regarding the hyperparameter resolution scale $R$, we have validated its impact on detection performance in the SMD dataset in Section 4.3. From this, we selected the recommended value $R=4$, which we then generalized to the other four datasets to report performance, as shown in Table 1. To more comprehensively evaluate the effectiveness of the multi-resolution setting and the sensitivity of hyperparameter $R$, we fixed the sampling steps at $L=20$ to sacrifice a small amount of accuracy in favor of inference efficiency, while adjusting $R$ on the other four datasets to observe performance changes. As shown in the table below, the optimal resolution scale $R$ varies slightly across different datasets. We attribute this variation to differences in anomaly rates and the types of anomaly patterns contained in each dataset, which require tuning $R$ to allow the model to effectively benefit from multi-resolution data. In general, for datasets with a low anomaly rate and many short-duration anomalies (lasting only a few time points), we recommend setting a lower resolution scale $R$ (e.g., 4 for the SMD and SMAP datasets) to ensure that anomalies are not overlooked in the low-resolution data. For datasets with a higher anomaly rate and longer-duration anomalies, a higher $R$ should be used to allow the model to fully leverage the temporal correlations across different resolution scales and capture both normal and anomalous event patterns.
>
> | Resolution Scale | | PSM | | | MSL | | | SWaT | | | SMAP | |
> |:-:|:-:|:-:|:-:|:-:|:-:|:-:|:-:|:-:|:-:|:-:|:-:|:-:|
> | $R$  | P | R | F1| P | R | F1| P | R | F1| P | R | F1| P | R | F1|
> |1 | 94.27 | 96.14 | 95.20 | 87.84 | 86.25 | 87.03 | 86.30 | 89.21 | 87.73 | 87.21 | 95.40 | 91.12 |
> |2 | 95.02 | 96.81 | 95.91 | 89.76 | 87.02 | 88.37 | 87.44 | 90.65 | 89.02 | 87.92 | 96.32 | 91.92 |
> |3 | 96.18 | 97.70 | 96.93 | 90.55 | 87.74 | 89.12 | 88.92 | 92.43 | 90.64 | 88.37 | 96.93 | 92.45 |
> |4 | 96.97 | 98.35 | 97.65 | 91.28 | 88.32 | 89.77 | 89.42 | 93.08 | 91.21 | 89.05 | 97.40 | **93.04** |
> |5 | 97.46 | 98.27 | 97.86 | 91.26 | 88.84 | **90.03** | 89.51 | 93.43 | **91.43** | 88.50 | 97.22 | 92.66|
> |6 | 97.83 | 98.23 | **98.03** | 90.92 | 88.41 | 89.65 | 89.30 | 93.49 | 91.34 | 88.41 | 97.38 | 92.68 |
> |7 | 97.12 | 97.96 | 97.54 | 88.84 | 87.75 | 88.29 | 88.75 | 92.51 | 90.59 | 87.22 | 96.17 | 91.47 |
> |8 | 96.27 | 97.41 | 96.83 | 88.05 | 86.87 | 87.46 | 86.23 | 90.40 | 88.57 | 86.53 | 95.49 | 90.78 |
>
> Secondly, as is well-known, there is a trade-off between precision and recall. As described in Appendix B.4, we can balance this trade-off by adjusting the voting threshold $\zeta$ during the ensemble inference process. Specifically, we ensure that the optimal resolution scale $R$ is maintained across all datasets, and then dynamically vary the value of $\zeta$ to observe the impact on anomaly detection performance. As shown in the table below, our approach does not solely focus on optimizing recall to improve detection performance. Regardless of whether $\zeta$ is increased or decreased (except when $\zeta$ is set to a very low value, such as $\zeta=2$), our model consistently maintains stable performance, with no significant drop in F1-score. In practical applications, by setting a higher value for $\zeta$, our model reduces false positives and thus achieves higher precision. Conversely, a lower $\zeta$ allows the model to detect more anomalies, leading to a higher recall. This enables a better balance between precision and recall, tailoring the model to meet different real-world requirements.
>
> | Voting Threshold | | SMD | | | SMAP | | | SWaT | |
> |:-:|:-:|:-:|:-:|:-:|:-:|:-:|:-:|:-:|:-:|
> | $\zeta$  | P | R | F1| P | R | F1| P | R | F1|
> |2 | 92.51 | 98.03 | 95.19 | 87.62 | 98.12 | 92.57 | 87.26 | 95.03 | 90.98 |
> |5 | 94.68 | 97.28 | 95.96 | 88.20 | 97.95 | 92.81 | 88.27 | 94.50 | 91.27 |
> |8 | 95.19 | 96.77 | 95.98 | 88.78 | 97.71 | 93.02 | 88.90 | 93.87 | 91.31 |
> |10| 95.70 | 96.32 | 96.01 | 89.05 | 97.40 | 93.04 | 89.51 | 93.43 | 91.43 |
> |13| 96.15 | 95.93 | 96.03 | 89.63 | 96.83 | 93.09 | 90.16 | 92.93 | 91.52 |
> |17| 96.82 | 95.08 | 95.94 | 90.07 | 96.29 | 93.07 | 90.94 | 92.01 | 91.47 |
> |20| 97.27 | 94.76 | 95.99 | 90.81 | 95.27 | 92.98 | 92.07 | 91.05 | 91.55 |
> |25| 97.62 | 94.18 | 95.86 | 92.10 | 93.70 | 92.89 | 92.95 | 90.08 | 91.49 |

---

> ### Author Response · Authors · 2024-11-27
> **Rebuttal by Authors 2**
>
> Further, in our frequency-enhanced decomposable network, we select frequency components with the top $m$ percent of amplitudes as stationary factors based on spectral statistics, with the remaining frequency components treated as non-stationary factors. As mentioned in Appendix B.6, we set the hyperparameter percentage $m$ to $20$, because the frequencies corresponding to the top $20\%$ of amplitudes account for over $90\%$ of all frequency components, which aligns with the reality where stationary factors are dominant. To investigate the impact of percentage $m$ on the detection performance further, we have conducted a sensitivity analysis on the SMD dataset regarding $m$ in the original manuscript, as illustrated in Table 9. For your convenience, we have listed the table and discussion below for your review.
>
> Variations within a reasonable range of $m$ do not cause drastic changes in performance, demonstrating that this frequency-based selection approach is robust. If $m$ is set too low (see $m=2$) or too high (see $m=50$), the model's performance significantly decreases because the time-variant and time-invariant variables are not effectively separated.
> | $m$ | Frequency Ratio | Precision | Recall | F1-score |
> |:-:|:-:|:-:|:-:|:-:|
> | 2 |  66.89% | 95.34 | 95.67 | 95.50 |
> | 10 |  85.33% | 95.68 | 96.10 | 96.88 |
> | 15 |  93.56% | 95.66 | 96.25 | 95.95 |
> | 20 |  91.45% | 95.70 | 96.32 | 96.01 |
> | 25 |  93.56% | 95.74 | 96.21 | 95.97 |
> | 50 |  98.73% | 95.40 | 95.76 | 95.57 |
>
> As shown in Figures 3 and 6, our time-variant and time-invariant encoders are primarily composed of the proposed Dilated ModernTCN Blocks (DMTB) and residual blocks, respectively. We next explore how varying the number of DMTBs and residual blocks affects anomaly detection performance, providing practical insights for fine-tuning network architecture.
>
> We begin by investigating the relationship between the number of DMTBs and detection performance, as illustrated in the table below. Keeping other hyperparameters constant, we start by setting the number of DMTBs to $0$, which means that only linear layers and hierarchical Transformers are used to model the non-stationarity. In this configuration, the model performs the worst on both datasets. As the number of DMTBs increases from $0$ to $4$, the model's detection performance steadily improves, reaching its highest point. Specifically, the F1-score on the SMD and SWaT datasets increased by $1.05\%$ (from $95.01$ to $96.01$) and $1.29\%$ (from $90.05$ to $91.21$), respectively. However, when the number of DMTBs is further increased beyond $4$, the performance no longer improves and instead slightly declines on both datasets. This suggests that the benefit of decoupling the time-invariant and time-variant components from complex non-stationary temporal patterns, based on spectral information, diminishes with excessive use of DMTBs. Overusing DMTBs may lead to overfitting, where normal non-stationarity is misidentified as anomalous behavior.
> | Number of DMTBs |   | SMD  |  |  | SWaT | |
> |:-:|:-:|:-:|:-:|:-:|:-:|:-:|
> | | P | R | F1| P | R | F1 | P | R | F1 |
> | 0  | 94.59  | 95.44  | 95.01  | 88.30  | 91.86  | 90.05   |
> | 1   | 94.82  | 95.75  | 95.28  | 88.82  | 92.45  | 90.60   |
> | 2  | 95.15  | 95.93  | 95.54  | 89.16  | 92.73  | 90.91   |
> | 3  | 95.48  | 96.21  | 95.84  | 89.30  | 92.94  | 91.08   |
> | 4  | 95.70  | 96.32  | 96.01  | 89.42  | 93.08  | 91.21   |
> | 5  | 95.64  | 96.28  | 95.95  | 89.33  | 93.16  | 91.20   |
> | 6  | 95.52  | 96.04  | 95.78  | 89.17  | 92.85  | 90.97   |
>
> The residual blocks in the time-invariant encoder are designed to capture global dynamic information. As illustrated in the table below, since time-invariant patterns tend to be simpler than time-varying components, we found that increasing the number of residual blocks to $2$ yielded satisfactory detection performance. Increasing the number beyond $2$ did not result in significant performance gains but did incur additional computational cost. Therefore, we decided to set the number of residual modules in the time-invariant encoder to $2$.
> | Number of Residual Blocks |   | SMD  |  |  | SWaT | |
> |:-:|:-:|:-:|:-:|:-:|:-:|:-:|
> | | P | R | F1| P | R | F1 | P | R | F1 |
> | 0 | 94.38  | 95.54  | 94.96  | 88.54  | 92.45  | 90.45   |
> | 1  | 95.37  | 96.16  | 95.76  | 88.97  | 93.01  | 90.94   |
> | 2  | 95.70  | 96.32  | 96.01  | 89.42  | 93.08  | 91.21   |
> | 3  | 95.62  | 96.26  | 95.94  | 89.56  | 92.98  | 91.23   |
> | 4  | 95.54  | 96.12  | 95.83  | 89.22  | 92.90  | 91.02   |
>
> The above experimental results and discussions have been included in Appendix B.9 of the updated manuscript, specifically in lines 1262-1358, highlighted in red.

---

> ### Author Response · Authors · 2024-11-27
> **Rebuttal by Authors 3**
>
> **W2: Clarification of resolution scales.**
>
> It seems there has been a misunderstanding regarding the resolution hyperparameter $R$. As described in Section 3.2, the multi-resolution data sampler, when $R=4$, applies an average pooling operation with a scale of $2^4=16$ on the highest-resolution data, resulting in low-resolution data that is compressed by a factor of $16$. This does not conflict with our main idea, which advocates exploring normal and anomalous temporal patterns across multiple resolution scales, allowing our model benefit from both general trend information provided by low-resolution data and the detailed event information from high-resolution data.
>
> To more comprehensively evaluate the effectiveness of the multi-resolution setting, we adjusted resolution scale $R$ on the other four datasets to observe performance changes. As shown in the table below, the optimal resolution scale $R$ varies slightly across different datasets. We attribute this variation to differences in anomaly rates and the types of anomaly patterns contained in each dataset, which require tuning $R$ to allow the model to effectively benefit from multi-resolution data. In general, for datasets with a low anomaly rate and many short-duration anomalies (lasting only a few time points), we recommend setting a lower resolution scale $R$ (e.g., 4 for the SMD and SMAP datasets) to ensure that anomalies are not overlooked in the low-resolution data. For datasets with a higher anomaly rate and longer-duration anomalies, a higher $R$ should be used to allow the model to fully leverage the temporal correlations across different resolution scales and accurately capture anomalous data.
>
> | Resolution Scale | | PSM | | | MSL | | | SWaT | | | SMAP | |
> |:-:|:-:|:-:|:-:|:-:|:-:|:-:|:-:|:-:|:-:|:-:|:-:|:-:|
> | $R$  | P | R | F1| P | R | F1| P | R | F1| P | R | F1| P | R | F1|
> |1 | 94.27 | 96.14 | 95.20 | 87.84 | 86.25 | 87.03 | 86.30 | 89.21 | 87.73 | 87.21 | 95.40 | 91.12 |
> |2 | 95.02 | 96.81 | 95.91 | 89.76 | 87.02 | 88.37 | 87.44 | 90.65 | 89.02 | 87.92 | 96.32 | 91.92 |
> |3 | 96.18 | 97.70 | 96.93 | 90.55 | 87.74 | 89.12 | 88.92 | 92.43 | 90.64 | 88.37 | 96.93 | 92.45 |
> |4 | 96.97 | 98.35 | 97.65 | 91.28 | 88.32 | 89.77 | 89.42 | 93.08 | 91.21 | 89.05 | 97.40 | **93.04** |
> |5 | 97.46 | 98.27 | 97.86 | 91.26 | 88.84 | **90.03** | 89.51 | 93.43 | **91.43** | 88.50 | 97.22 | 92.66|
> |6 | 97.83 | 98.23 | **98.03** | 90.92 | 88.41 | 89.65 | 89.30 | 93.49 | 91.34 | 88.41 | 97.38 | 92.68 |
> |7 | 97.12 | 97.96 | 97.54 | 88.84 | 87.75 | 88.29 | 88.75 | 92.51 | 90.59 | 87.22 | 96.17 | 91.47 |
> |8 | 96.27 | 97.41 | 96.83 | 88.05 | 86.87 | 87.46 | 86.23 | 90.40 | 88.57 | 86.53 | 95.49 | 90.78 |
>
>
> **W3: Clarification on the precision-recall tradeoffs.**
>
> It is well-known that there is an inherent trade-off between precision and recall. As described in Appendix B.4, we can balance this trade-off by adjusting the voting threshold $\zeta$ during the ensemble inference process. The performance improvement of our method, particularly in recall, is mainly due to setting a relatively small voting threshold $\zeta=10$, which allows more anomalies to be detected with a more lenient criterion. To verify that our model does not simply optimize recall to enhance detection performance, we varied the voting threshold $\zeta$ across three datasets and observed its impact on anomaly detection performance. As shown in the table below, as $\zeta$ increases, our model reduces false positives and achieves higher precision. However, this also leads to a decrease in recall. Despite this trade-off, our model continues to demonstrate stable performance, as evidenced by only slight fluctuations in F1-score. This indicates that our model effectively balances precision and recall to meet the practical requirements of real-world anomaly detection tasks.
>
> | Voting Threshold | | SMD ($R=4$) | | | SMAP ($R=4$) | | | SWaT ($R=5$) | |
> |:-:|:-:|:-:|:-:|:-:|:-:|:-:|:-:|:-:|:-:|
> | $\zeta$  | P | R | F1| P | R | F1| P | R | F1|
> |2 | 92.51 | 98.03 | 95.19 | 87.62 | 98.12 | 92.57 | 87.26 | 95.03 | 90.98 |
> |5 | 94.68 | 97.28 | 95.96 | 88.20 | 97.95 | 92.81 | 88.27 | 94.50 | 91.27 |
> |8 | 95.19 | 96.77 | 95.98 | 88.78 | 97.71 | 93.02 | 88.90 | 93.87 | 91.31 |
> |10| 95.70 | 96.32 | 96.01 | 89.05 | 97.40 | 93.04 | 89.51 | 93.43 | 91.43 |
> |13| 96.15 | 95.93 | 96.03 | 89.63 | 96.83 | 93.09 | 90.16 | 92.93 | 91.52 |
> |17| 96.82 | 95.08 | 95.94 | 90.07 | 96.29 | 93.07 | 90.94 | 92.01 | 91.47 |
> |20| 97.27 | 94.76 | 95.99 | 90.81 | 95.27 | 92.98 | 92.07 | 91.05 | 91.55 |
> |25| 97.62 | 94.18 | 95.86 | 92.10 | 93.70 | 92.89 | 92.95 | 90.08 | 91.49 |

---

> ### Author Response · Authors · 2024-11-27
> **Rebuttal by Authors 4**
>
> **W4: Details on the pooling operation in the MRD-sampler.**
>
> To provide a clearer explanation of the MRD-Sampler, we have revised its description in the updated manuscript as follows (see lines 197-201 in Section 3.2, highlighted in red):
>
> "Given a time series $x_0$ and a resolution scale $r \in [1, R]$, this step employs average pooling (Wu et al., 2021) to generate multi-resolution data, $x ^ r _ 0$, that is, $x ^ r _ 0 = \text{AvgPool}(\text{Padding}(x_0), 2^r )$, where $2^r$ denotes the pooling size that increases with $r$. $\text{AvgPool(·)}$ computes the average of every $2^r$ non-overlapping points. $\text{Padding(·)}$ is then applied by replicating the pooled data at $r$ resolution $2^r$ times to maintain the same length as the highest-resolution series $x_0$."
>
>
> **W5: Statement on the code availability.**
>
> We assure that all experimental results are reproducible, and we will release our code and pretrained weights upon acceptance to facilitate further research within the community.
>
> We hope the above responses have adequately addressed your concerns, and we would greatly appreciate it if you could adjust your rating accordingly. If you still have any remaining questions or concerns about our work, please do not hesitate to let us know, and we will respond promptly to resolve them.
>
>
>
> Best regards,
>
> Authors

---

> ### Author Response · Authors · 2024-11-30
> **Request for Reviewer's Attention and Feedback**
>
> Dear Reviewer cQvr:
>
> We would like to express our sincere gratitude for the time and effort you dedicated to reviewing our manuscript, as well as for your insightful and constructive comments. We have carefully considered each of your comments and have provided detailed responses to address the concerns you raised.
>
> As the reviewer-author discussion phase is scheduled to conclude in less than three days, we kindly request that you confirm whether our responses have sufficiently addressed your concerns, and if there are any remaining issues that still require attention. We greatly appreciate your feedback and would be grateful for any further suggestions you may have.
>
> We look forward to your feedback and thank you once again for your continued support. We hope you have a wonderful and restful Thanksgiving holiday!
>
> Best regards,
>
> Authors

---

> ### Author Response · Authors · 2024-12-02
> **Request for Reviewer's Attention and Feedback**
>
> Dear Reviewer cQvr:
>
> Once again, we sincerely appreciate the time and effort you dedicated to reviewing our manuscript, as well as your insightful comments.
>
> As the discussion period is drawing to a close, we would like to bring your attention, and kindly request that you confirm whether our responses have adequately addressed your concerns. If so, we would be grateful if you could consider adjusting the rating accordingly.
>
> If you have any further suggestions or questions, please do not hesitate to let us know, and we will promptly provide our responses. We eagerly look forward to your feedback and deeply appreciate your continued support.
>
> Best regards,
>
> Authors

---

> > ### Comment · Reviewer_cQvr · 2024-12-02
> >
> > The authors have provided clear and extensive comments and additional experiments to address my initial concerns. I believe the additional experiments and modifications have resulted in a significantly improved manuscript and I have changed my score accordingly.

---

> > > ### Author Response · Authors · 2024-12-02
> > >
> > > We sincerely appreciate your constructive pre-rebuttal review and your decision to raise the rating.
> > >
> > > Should you have any remaining concerns or unresolved questions regarding our work, please feel free to inform us. Your valuable feedback will greatly assist us in further improving our manuscript.
> > >
> > > Thank you once again for your time and insightful comments.

---

### Comment · Area_Chair_zR9f · 2024-11-25
**Acknowledge the author responses**

Dear Reviewers,

Thank you very much for your effort. As the discussion period is coming to an end, please acknowledge the author responses and adjust the rating if necessary.

Sincerely,
AC

---

> ### Author Response · Authors · 2024-11-25
>
> Dear Area Chair and Reviewers,
>
> We sincerely thank the reviewers for their insightful comments and constructive suggestions, as well as the AC for the attention and reminders. **We apologize for the delay in our response**, as additional time was needed to complete further experiments and revise the manuscript.
>
> **We kindly ask for your patience as we finalize our responses, which will be released shortly**.
>
> Best regards,
>
> Authors

---

### Author Response · Authors · 2024-11-27
**Global Response**

# Global Response

We extend our sincere gratitude to all the reviewers (**R1**-cQvr, **R2**-meLm, **R3**-hGSS, **R4**-VctK) for their insightful comments and constructive suggestions, which have greatly helped us emphasize the contributions of our approach and improve the quality of our paper. We are encouraged to hear that all reviewers acknowledged our paper as well-organized and clearly presented (**R1**, **R2**, **R3**, **R4**), with great motivations (**R2, R4**) and comprehensive experimental evaluations (**R1, R3**).

We sincerely apologize for the delay in our response, as additional time was required to conduct further experiments and revise the manuscript. In response to reviewers’ comments, we have systematically addressed each point in individual replies. Below, we provide an overview of the revisions made:

+ We have refined certain statements and analyses of experimental results to improve clarity and readability.

+ We have discussed more outstanding previous works to thoroughly clarify the contributions of our work.

+ We have conducted additional experimental results on hyperparameter sensitivity, precision-recall tradeoffs, and computational efficiency analysis.

Thanks again for the reviewers’ valuable suggestions. We have updated the paper accordingly (using four colors to highlight the revisions for the reviewers’ convenience during the review process), and the code will be released upon acceptance for community research. We deeply appreciate the reviewers' time and effort in checking our responses and hope that our clarifications have adequately addressed all concerns.

Should there be any remaining uncertainties or questions regarding our work, please do not hesitate to let us know, and we will respond promptly to address them.

---

### Comment · Area_Chair_zR9f · 2024-11-28
**Discussion needed**

Dear Reviewers,

As you are aware, the discussion period has been extended until December 2. Therefore, I strongly urge you to participate in the discussion as soon as possible if you have not yet had the opportunity to read the authors' response and engage in a discussion with them. Thank you very much.

Sincerely,
Area Chair

---

### Meta-Review · Area_Chair_zR9f · 2024-12-19

**Metareview:**

This paper presents an anomaly detection algorithm for non-stationary time series, which integrates a coarse-to-fine diffusion paradigm with a frequency-enhanced decomposable network.  All the reviewers found that the paper is well-written and interesting, they also raised several concerns mainly on the evaluation (hyperparameter setting and computational cost) and novelty of the proposed model.  The authors successfully addressed many of such concerns during the discussion period.  Regarding the novelty issue, I admit that sophisticated optimization based on existing techniques can be a meaningful contribution.  All the reviewers are favorable on the acceptance of this paper.  Thus, I would like to recommend an accept.

**Additional Comments On Reviewer Discussion:**

All the reviewers were satisfied with the authors' responses.  The authors provided very extensive results based on further experiments.

---

### Decision · Program_Chairs · 2025-01-22

Accept (Poster)